# NETFORMER: AN INTERPRETABLE MODEL FOR RECOVERING DYNAMICAL CONNECTIVITY IN NEURONAL POPULATION DYNAMICS

**Ziyu Lu**[1*], **Wuwei Zhang**[2*], **Trung Le**[1],
**Hao Wang**[3], **Uygar Sümbül**[4], **Eric Shea-Brown**[1†], **Lu Mi**[5†]
[1]University of Washington    [2]Princeton University    [3]Rutgers University
[4]Allen Institute for Brain Science    [5]Georgia Institute of Technology

## ABSTRACT

Neuronal dynamics are highly nonlinear and nonstationary. Traditional methods for extracting the underlying network structure from neuronal activity recordings mainly concentrate on modeling static connectivity, without accounting for key nonstationary aspects of biological neural systems, such as ongoing synaptic plasticity and neuronal modulation. To bridge this gap, we introduce the NetFormer model, an interpretable approach applicable to such systems. In NetFormer, the activity of each neuron across a series of historical time steps is defined as a token. These tokens are then linearly mapped through a query and key mechanism to generate a state- (and hence time-) dependent attention matrix that directly encodes nonstationary connectivity structures. We analyze our formulation from the perspective of nonstationary and nonlinear networked dynamical systems, and show both via an analytical expansion and targeted simulations how it can approximate the underlying ground truth. Next, we demonstrate NetFormer's ability to model a key feature of biological networks, spike-timing-dependent plasticity, whereby connection strengths continually change in response to local activity patterns. We further demonstrate that NetFormer can capture task-induced connectivity patterns on activity generated by task-trained recurrent neural networks. Thus informed, we apply NetFormer to a multi-modal dataset of real neural recordings, which contains neural activity, cell type, and behavioral state information. We show that NetFormer effectively predicts neural dynamics and identifies cell-type specific, state-dependent dynamic connectivity that matches patterns measured in separate ground-truth physiology experiments, demonstrating its ability to help decode complex neural interactions based on population activity observations alone.

## 1 INTRODUCTION

Inferring the underlying connectivity of a network from observations of the activity of its units is a long-standing challenge. In the brain, this challenge is exacerbated by (i) different nonlinear dynamics present in individual neurons, (ii) the difficulty of experimentally sampling the full neuronal population simultaneously, and (iii) dynamic reconfiguration of effective connectivity, mediated by both synaptic plasticity and neuromodulation. This last issue carries significant practical importance in studying behavioral dynamics, learning and memory (Bargmann, 2012; Tyulmankov et al., 2021; Marder, 2012; Liu et al., 2021; Aitken & Mihalas, 2023). As such, it poses a (harder) generalization of the classical problem where the connectivity should no longer be considered as a static unknown, rather as a dynamical variable that needs to be inferred and tracked over time.

A surrogate, but not sufficient, measure of success in unsupervised inference of connectivity is the inferred network's success in fitting the observed dynamics. While traditional linear dynamical

---

*Equal contribution (co-first authors).

†Equal contribution (co-senior authors).

Correspondence    to:    `luziyu@uw.edu, wz1411@princeton.edu, etsb@uw.edu, lmi7@gatech.edu`.

models struggle to capture the essential nonlinear mechanisms of leaky integration and firing (Gerstner & Kistler, 2002) in biological neurons, more sophisticated nonlinear models typically suffer from a lack of interpretability, making it difficult to identify the underlying connectivity (Pandarinath et al., 2018; Le & Shlizerman, 2022; Ye et al., 2023). Moreover, traditional approaches often adopt a static perspective on connectivity (Tank et al., 2021; Löwe et al., 2022), failing to account for the nonstationary interactions, such as those produced by plasticity and modulation at synapses.

Here we propose an interpretable nonlinear and nonstationary dynamical model to represent interactions between neurons (Figure 1), based on the fast weight programming nature of the attention mechanism (Schlag et al., 2021). Prior research has suggested that the attention mechanism can reveal information about the underlying structure of a system (Singh & Buckley, 2023; Lu et al., 2023). We further removed the softmax activation function in the attention mechanism, as the constraint of attention weights summing up to one is not biologically meaningful because neither the in-degrees nor the out-degrees of neuronal connectivity (nor their counterparts incorporating synaptic strength) are invariant across neurons (Santuy et al., 2020). We first demonstrated with both mathematical analysis and simulation study that even without the softmax activation, the core part of the attention mechanism – the dot-product between queries and keys – is capable of capturing nonstationary and nonlinear structural information. Next, we applied this novel approach to a wide range of simulated networks including nonstationary and/or complex nonlinear connectivity patterns, and showed that it can recover ground truth connectivity information. We then applied it to a large-scale, publicly available dataset of neuronal activity recordings. Importantly, this dataset includes the genetic *cell type* of individual neurons, enabling us to compare our predictions for cell-type level connectivity patterns against known ground truth values from independent experiments. Taken together, this shows the potential of our method for recovering interpretable connectivity information, even in the presence of complex nonlinear and nonstationary network dynamics.

Our main contributions are as follows: (i) We formulated a trans**former**-inspired **net**work model, the NetFormer, for which the core of the attention mechanism – the dot-product between queries and keys – directly encodes nonstationary and nonlinear structure of networks; (ii) On a simulated network with the spike-timing-dependent plasticity mechanism, we demonstrated that the inferred time-varying weights from attention aligned with the underlying changes in connectivity; (iii) On activity generated by task-trained recurrent neural networks, we demonstrated that attention can capture task-induced connectivity patterns; (iv) We applied the NetFormer model to population activity recorded from mouse visual cortex, and showed that attention can recover experimentally measured synaptic connectivity, while benchmarking it with standard recurrent models and other common statistical metrics. Additionally, we demonstrated that attention can naturally reflect state-dependent modulations in the inferred cell-type level connectivity, even more effectively compared to two other specialized baseline methods.

## 1.1 RELATED WORK

**Dynamical models of neuronal activity.** Dynamical models have been a powerful tool for high-dimensional neural data analysis (Vyas et al., 2020). Generalized linear models (GLMs), known for both interpretability and desirable convexity properties (Paninski et al., 2007), have been widely used to model neuronal population activity as well as inter-neuronal interactions (Pillow et al., 2008; Das & Fiete, 2020). Nevertheless, unless stacked with an explicit state switching mechanism (Escola et al., 2011), in GLMs the weight matrix describing interactions among neurons is typically stationary across time (Li et al., 2024). Recurrent neural networks (RNNs) have been a popular alternative (Barak, 2017; Perich et al., 2020); while the connectivity in (trained) RNNs is typically given by a static connectivity matrix "$W$", variants including long short-term memory networks (LSTMs) (Hochreiter & Schmidhuber, 1996) and gated recurrent neural networks (GRUs) (Cho et al., 2014), do include nonstationarities at the level of individual neural units. While this can enhance the model's expressivity and performance in predictive tasks (Salinas et al., 2020; Lai et al., 2018), it also introduces challenges for interpretability (Tank et al., 2021). Recently, transformer models (Vaswani et al., 2017) have been observed to outperform RNNs in various time series forecasting tasks (Zhou et al., 2021; Wu et al., 2021), but their deep layered structures and nonlinear attention mechanisms also raise challenges in interpretation with respect to underlying connectivity structures in the original data (Jain & Wallace, 2019; Abnar & Zuidema, 2020), as discussed more below. A closely related approach to the present work is the switching linear dynamical systems (Fox et al., 2008; Linderman et al., 2017a). These models have nonstationary connectivity matrices which switch among a number of discrete values

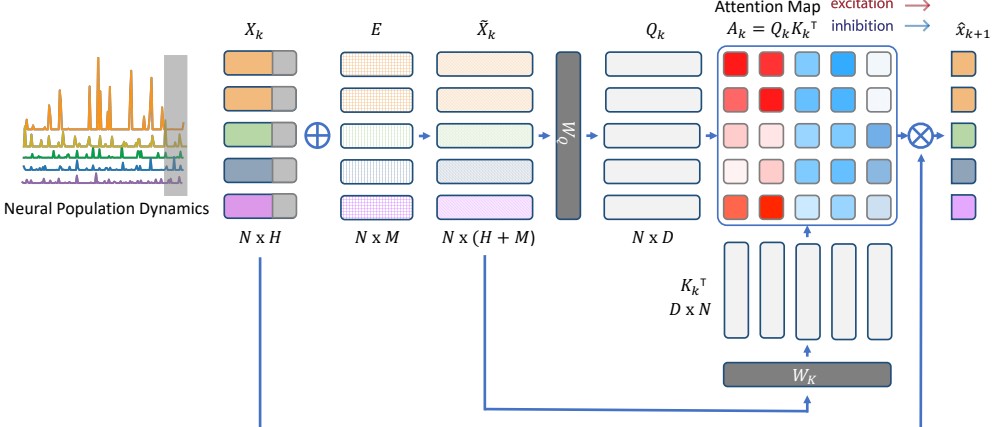

**Figure 1: Overview of NetFormer**. NetFormer learns to predict neural dynamics and infer dynamical connectivity through a linearized attention mechanism. The model takes in activities $\boldsymbol{X}_k$ of $N$ neurons across $H$ history timesteps, and predicts their activity $\boldsymbol{x}_{k+1}$ at timestep $k+1$. Queries $\boldsymbol{Q}_k$ and keys $\boldsymbol{K}_k$ are linearly mapped from $\tilde{\boldsymbol{X}}_k$ ($\boldsymbol{X}_k$ concatenated with positional embedding $\boldsymbol{E}$), using $\boldsymbol{W}_Q$ and $\boldsymbol{W}_K$. The time-dependent linearized attention matrix $\boldsymbol{A}_k$ is computed as $\boldsymbol{Q}_k\boldsymbol{K}_k^\top$, which learns the neuron-level dynamical connectivity. Red and blue colors in the attention map indicate excitatory and inhibitory interactions, respectively.

according to a Markov process. Nevertheless, in vivo experiments have revealed that cortical activity is more likely to go through a continuum of states instead of discrete switching (Harris & Thiele, 2011). This motivates us to propose a model capable of capturing continuous changes in connectivity.

**Interpretability of the attention mechanism.** Attention weights and positional embeddings provide opportunities to understand the inner working of the transformer models. However, the interpretability of these components is still a subject of debate. Findings supporting a certain level of interpretability, such as correlation to linguistic features, are common in the literature, with specialized metrics developed to quantify their interpretability (Clark et al., 2019; Abnar & Zuidema, 2020). However, caution should be taken when equating attention with explanation (Jain & Wallace, 2019), considering the lack of identifiability (Brunner et al., 2019) and the wide variety of underlying architectures and implementations (Wang & Chen, 2020). In this work, we seek to avoid these confounding aspects by focusing on the linearized attention mechanism (Schlag et al., 2021).

**Predicting activity from connectivity.** The reverse direction, predicting activity from connectivity, is an allied approach for studying the complexities relating functional and structural information. A prominent line of study has focused on the worm C. elegans as its synaptic connectome was the first available among all species (White et al., 1986). Using this, generative models of activity have been proposed (Mi et al., 2021). Nevertheless, decades of electrophysiological analyses have emphasized the strong additional role of neuromodulators in shaping activity (Randi et al., 2023; Marder, 2012). As a result, the synaptic connectome alone predicts only partial information about recorded population dynamics (Bargmann, 2012; Randi & Leifer, 2020).

## 2 AN INTERPRETABLE MODEL FOR RECOVERING DYNAMIC CONNECTIVITY

We consider an $N$-dimensional dynamical system

$$\frac{d}{dt}\boldsymbol{x}(t) = f\big(\boldsymbol{W}(t)\boldsymbol{x}(t)\big) \tag{1}$$

where $\boldsymbol{x}(t) \in \mathbb{R}^N$, $f : \mathbb{R}^N \to \mathbb{R}^N$, and $\boldsymbol{W}(t)$ is an $N \times N$ matrix whose entries may vary across time. $W_{i,j}(t)$ prescribes how the $i$-th variable $\boldsymbol{x}^{(i)}$ is driven by the $j$-th variable $\boldsymbol{x}^{(j)}$ at time $t$.

Let $\boldsymbol{x}_k$ be observations of the system at discrete timesteps $t_k$. For each $k$, we train the NetFormer model (Figure 1) to predict $\boldsymbol{x}_{k+1}$ based on $\boldsymbol{X}_k = [\boldsymbol{x}_{k-H+1} \quad \cdots \quad \boldsymbol{x}_k] \in \mathbb{R}^{N \times H}$, the recent $H$-step history of the system up to timestep $k$. To encode neuronal identities, a learnable positional embedding matrix $\boldsymbol{E} \in \mathbb{R}^{N \times M}$ is concatenated to $\boldsymbol{X}_k$, giving $\tilde{\boldsymbol{X}}_k = [\boldsymbol{X}_k \quad \boldsymbol{E}] \in \mathbb{R}^{N \times (H+M)}$. The queries $\boldsymbol{Q}_k$ and keys $\boldsymbol{K}_k$ are obtained through linear transformations of $\tilde{\boldsymbol{X}}_k$, and their product gives the linearized attention matrix $\boldsymbol{A}_k$:

$$\boldsymbol{Q}_k = \tilde{\boldsymbol{X}}_k\boldsymbol{W}_Q \in \mathbb{R}^{N \times D}, \ \boldsymbol{K}_k = \tilde{\boldsymbol{X}}_k\boldsymbol{W}_K \in \mathbb{R}^{N \times D}, \ \boldsymbol{A}_k = \boldsymbol{Q}_k\boldsymbol{K}_k^\top \in \mathbb{R}^{N \times N}. \tag{2}$$

It follows that entry $(i, j)$ of $\boldsymbol{A}_k$ is computed from the history of $\boldsymbol{x}^{(i)}$ and $\boldsymbol{x}^{(j)}$, and thus describes the relationship between the $i$-th and $j$-th variables. To predict $\boldsymbol{x}_{k+1}$, we take $\boldsymbol{x}_k$ to be the values $\boldsymbol{v}_k$

and employ the residual connection (He et al., 2016), obtaining prediction as

$$\hat{\boldsymbol{x}}_{k+1} = \boldsymbol{v}_k + \boldsymbol{A}_k \boldsymbol{v}_k = \boldsymbol{x}_k + \boldsymbol{A}_k \boldsymbol{x}_k, \tag{3}$$

which is similar to the update rule we would get if Equation 1 were simulated using the classical forward Euler method with step size $\delta$

$$\boldsymbol{x}_{k+1} = \boldsymbol{x}_k + \delta f(\boldsymbol{W}_k \boldsymbol{x}_k). \tag{4}$$

Since neuronal connections can be either excitatory or inhibitory, but neither effect can be arbitrarily large, we choose $f$ to be a sigmoidal function with

$$f(0) = 0, \ f(\xi) = f(0) + f'(0)\xi + O(\xi^3) \text{ for } \xi \text{ within some interval } (-\epsilon, \epsilon) \text{ around } 0^{[1]}$$

Equation 4 can thus be written as

$$\boldsymbol{x}_{k+1} = \boldsymbol{x}_k + \delta f(\boldsymbol{W}_k \boldsymbol{x}_k) = \boldsymbol{x}_k + \delta f'(0)\boldsymbol{W}_k \boldsymbol{x}_k + \delta O(\boldsymbol{x}_k^3). \tag{5}$$

Comparing equations 3 and 5, we deduce that the linearized attention matrix $\boldsymbol{A}_k$ learned by the NetFormer may capture the true interactions between different variables $\boldsymbol{W}_k$ by approximating $\delta f'(0)\boldsymbol{W}_k$, especially when $\epsilon < 1$ and the first order term plays the most significant role. It is not hard to see that this hypothesis also extends to systems in the form of

$$\frac{d}{dt}\boldsymbol{x}(t) = -\boldsymbol{x}(t) + f\big(\boldsymbol{W}(t)\boldsymbol{x}(t)\big), \tag{6}$$

which includes the decaying effect that is commonly present in neural dynamics (Gerstner & Kistler, 2002) (Appendix A.1). We provide empirical evidence for this hypothesis on both forms of systems in the subsequent sections. Our code is at `https://github.com/NeuroAIHub/NetFormer`.

## 3 EXPERIMENTS ON SYNTHETIC DATA

### 3.1 NONLINEAR AND NONSTATIONARY SYSTEMS SIMULATION

We first considered four simplified simulated systems, with variations in the inclusion of nonlinearity and nonstationarity:

$$\textbf{(a) } \frac{d\boldsymbol{x}}{dt} = \boldsymbol{W}\boldsymbol{x}, \ \textbf{(b) } \frac{d\boldsymbol{x}}{dt} = \tanh(\boldsymbol{W}\boldsymbol{x}), \ \textbf{(c) } \frac{d\boldsymbol{x}}{dt} = \boldsymbol{W}(\boldsymbol{x})\boldsymbol{x}, \ \textbf{(d) } \frac{d\boldsymbol{x}}{dt} = \tanh(\boldsymbol{W}(\boldsymbol{x})\boldsymbol{x}),$$

where $\boldsymbol{W}(\boldsymbol{x}) = \boldsymbol{W}_0 + \boldsymbol{x}\boldsymbol{\omega}^\top$. Simulation details are in Appendix A.2.1. All trained NetFormer models are able to make accurate one-step-ahead predictions ($R^2 = 1.000$, Figure 2a-d left). Visually, the average linearized attention matrix across timesteps, $\bar{\boldsymbol{A}} = \frac{1}{K}\sum_{k=1}^{K}\boldsymbol{A}_k$, provides a good characterization of the average ground-truth dynamical association matrix across timesteps, $\bar{\boldsymbol{W}} = \frac{1}{K}\sum_{k=1}^{K}\boldsymbol{W}(\boldsymbol{x}_k)$ (Figure 2a-d right). As a baseline, we consider $\boldsymbol{A}_{\text{OLS}}$ from the linear ordinary least squares regression $\hat{\boldsymbol{x}}_{k+1} = \boldsymbol{A}_{\text{OLS}}\boldsymbol{x}_k$. We used the Spearman's rank correlation coefficient ($\rho$) between the off-diagonal entries of $\bar{\boldsymbol{A}}$ or $\boldsymbol{A}_{\text{OLS}}$ and $\bar{\boldsymbol{W}}$ to quantify how faithfully the learned connectivities reflect the ground-truth. $\bar{\boldsymbol{A}}$ achieved comparable performance as $\boldsymbol{A}_{\text{OLS}}$ in systems **(a) (b)**, but significantly outperformed $\boldsymbol{A}_{\text{OLS}}$ in systems **(c) (d)**, both visually (Figure 2a-d right) and quantitively (Appendix A.2.2). Moreover, in the nonstationary systems **(c) (d)**, the linearized attention matrix is able to track the majority of changes in $\boldsymbol{W}(\boldsymbol{x})$ across timesteps (Figure 2e-f and Appendix A.2.3). This ability to capture nonstationarity also explains why NetFormer can outperform the linear regression model which only accounts for static connectivity.

### 3.2 SPIKE-TIMING-DEPENDENT PLASTICITY (STDP) SIMULATION

Next, we tested NetFormer in a more neurobiological realistic setting by considering a leaky integrate-and-fire (LIF) neuron (Gerstner & Kistler, 2002) with spike-timing-dependent plasticity (STDP). STDP is a fundamental and widely studied synaptic modification scheme in neuroscience (Bi & Poo, 1998; Abbott & Nelson, 2000; Gerstner et al., 1996; Song et al., 2000) where the synaptic connection strength between two neurons depends on the relative timing of the spikes they fire. In

---

[1] see Appendix A.2.4 for more discussion on the radius of convergence of this series representation

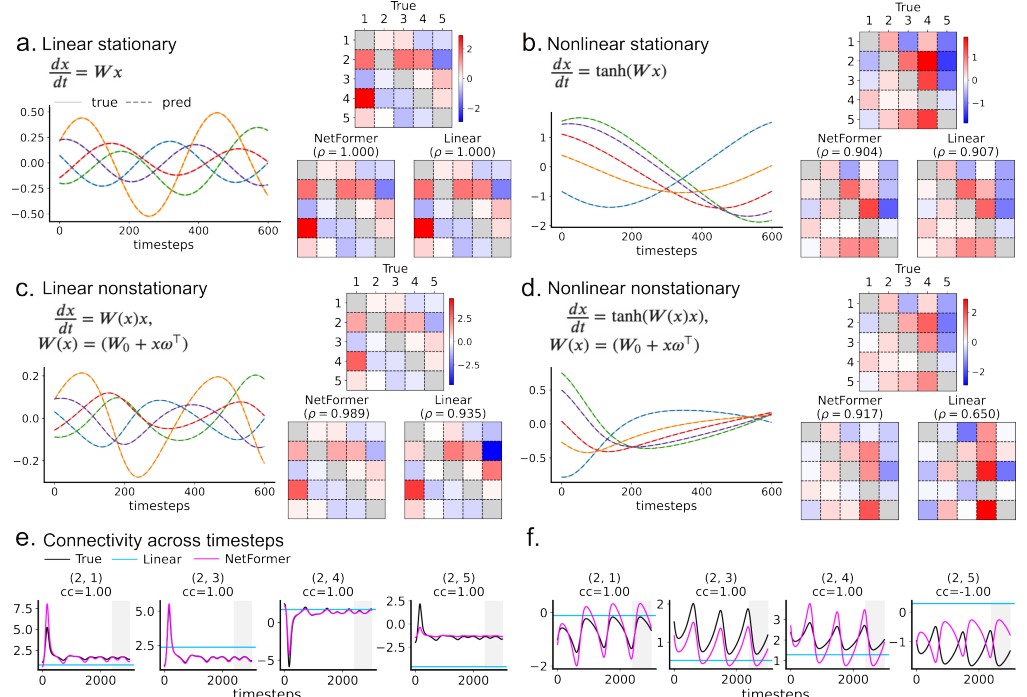

**Figure 2: NetFormer provides accurate dynamics predictions, and recovers ground truth connectivity matrices. a-d Left:** Test set predictions of NetFormer. Predicted trajectories were obtained by concatenating all one-step-ahead predictions. Predicted (dashed) trajectories overlap with the true (solid) trajectories. **a-d Right:** True and inferred connectivity from NetFormer's linearized attention matrix or linear regression weight matrix. For systems **c, d**, connectivity is averaged across test set timesteps for visualization. For NetFormer, the inferred connectivity shown is the one whose Spearman correlation $\rho$ is closest to the average $\rho$ across 10 random seeds. Colorbars: scale of off-diagonal entries. Diagonal entries are masked in grey. Inferred connectivity matrices were rescaled by the reciprocal of simulation stepsize for visualization. **e:** True and inferred temporal evolution of four example connections in the nonstationary system **c**. Timesteps used as test set are shaded in grey. **f:** True and inferred temporal evolution in system **d**.

typical models, when the presynaptic neuron fires before the postsynaptic neuron, their connection strengthens, increasing the postsynaptic response to future spikes. Conversely, presynaptic firing after postsynaptic weakens the connection. The amount of change in synaptic strength depends on the time interval between pre- and postsynaptic spikes. An example relationship is illustrated in Figure 3a.

We simulated a postsynaptic LIF neuron receiving excitatory inputs from 100 presynaptic neurons, following Neuromatch Academy (2023) (see Appendix A.3 for simulation details). Since we only need the spike times of presynaptic neurons, instead of simulating their dynamics, we directly modeled their spike trains with independent Poisson processes (Figure 3b). When a spike arrives at a certain synapse, the membrane potential of the LIF neuron is increased by an amount proportional to the synaptic strength, and this potential is reset once the firing threshold is reached (see Appendix A.3 for precise equations). The strength, or weight, of each synapse is modulated following STDP (Figure 3a) within some boundaries. The weight evolution of ten example synapses across the simulation timespan is shown in Figure 3c.

We trained NetFormer to predict the next-step membrane potential of the LIF neuron based on its present and past potential and the spikes it received. This relationship is captured by the $1 \times 101$ linearized attention matrix of NetFormer. After training, NetFormer is able to capture the dynamics well (test set MSE=$0.055 \pm 0.008$, $R^2$=$0.912 \pm 0.013$, mean $\pm$ std across 5 random seeds), outperforming the linear regression model (test set MSE=$0.138$, $R^2$=$0.777$). Visualization of their predictions (Figure 3d) shows that NetFormer effectively captures the nonlinear reset mechanism, whereas the linear regression model fails.

We further extracted the learned pre-to-postsynaptic neuron relationship from the linearized attention matrix of NetFormer and compared it against the ground-truth synaptic weights. Unlike the toy systems in Section 3.1 (Figure 2e, f), recovering individual synaptic weights at each timestep from the attention scores does not seem viable. This is not surprising, though, given that the weight differences between synapses are much smaller compared to the membrane potential, that a synapse is only involved in dynamics prediction when it has a spike, and the strong nonlinearity in the membrane spiking dynamics. Nevertheless, we found that for most synapses, the long-term trends in

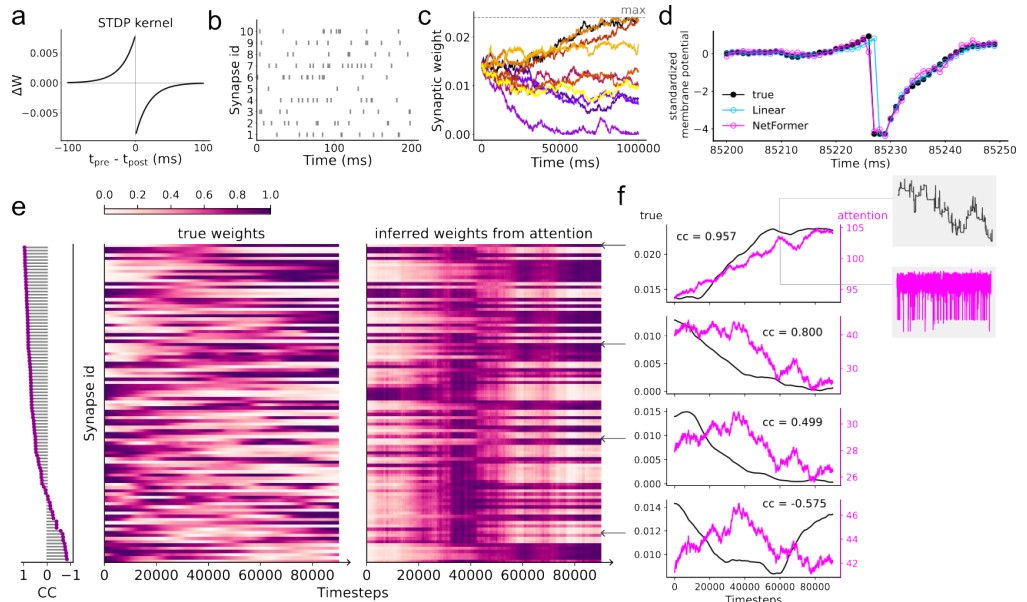

**Figure 3: NetFormer captures the effect of spike-timing-dependent plasticity (STDP). a.** STDP temporal kernel used in the simulation. Change in the synaptic weight ($\Delta W$) is a function of the delay between pre- and post-synaptic neuron spikes ($t_{pre} - t_{post}$). **b.** Raster plot of spikes received by the LIF neuron at the first 10 synapses during the first 200 timesteps. **c.** Evolution of the first 10 synapses' weights across the simulated timespan. Each color represents a synapse. **d.** True and predicted membrane potential across 100 timesteps in the test set. Membrane potential was z-scored before all model fitting. Predicted traces were constructed by concatenating one-step-ahead predictions. **e.** True and inferred synaptic evolution across simulation time span, after smoothing. Both true and inferred evolution trajectories were smoothed with a sliding window. Each row represents a synapse, and the rows were sorted by the correlation between the smoothed true and inferred trajectories. The correlation coefficients corresponding to each row are shown on the left. For visualization, each row was rescaled to $[0, 1]$ through min-max normalization. **f.** Weight evolution trajectories of four example synapses, which correspond to the 1st, 31st, 61st, and 91st rows from **e.**, as indicated by arrows. The trajectories were smoothed but not min-max normalized. Insets: Raw, unsmoothed trajectories in a sliding window.

the corresponding attention scores across timesteps are consistent with the true trends in synaptic weights driven by STDP (Figure 3e, f). Across the 100 synapses, the median of the correlation coefficients between smoothed synaptic weight evolution trajectories inferred from attention and the true weight trajectories is $0.608 \pm 0.028$ (mean $\pm$ std across 5 seeds). This demonstrates that the simple structure of NetFormer is capable of capturing nonstationary connectivity in spiking neuronal networks with dynamic synapses.

### 3.3 Task-driven Population Activity Simulation

We further examined whether NetFormer can identify task-driven connectivity patterns in neural populations. As single neuron level connectivity in task-performing laboratory animals are hard to measure, we resorted to task-trained recurrent neural network (RNN) models. The RNN models are trained to perform tasks which mirror the ones laboratory animals are trained to perform, and the activity of their recurrent hidden units has been widely adopted in studies of task-driven neural representation and computation (Mante et al., 2013; Sussillo et al., 2015; Yang et al., 2019; Duncker et al., 2020). Here we considered three representative tasks from the NeuroGym toolkit (Molano-Mazon et al., 2022): **a.** Perceptual Decision Making (Britten et al., 1992), **b.** Go-Nogo (Zhang et al., 2019), and **c.** Delay Comparision (Barak et al., 2010). For each task, we trained a RNN model with hidden dynamics

$$\boldsymbol{h}_k = \tanh(\boldsymbol{W}_{stim}\boldsymbol{s}_k + \boldsymbol{b}_{stim} + \boldsymbol{W}_{rec}\boldsymbol{h}_{k-1} + \boldsymbol{b}_{rec}). \tag{7}$$

Here $\boldsymbol{h}_k \in \mathbb{R}^N$ denotes the activity of hidden units, while $\boldsymbol{s}_k \in \mathbb{R}^S$ represents the stimulus input at timestep $k$. $\boldsymbol{W}_{rec} \in \mathbb{R}^{N \times N}$ specifies how current activity is shaped by past activity, and $\boldsymbol{W}_{stim} \in \mathbb{R}^{N \times S}$ captures the effect of the present stimulus input. $\boldsymbol{b}_{rec} \in \mathbb{R}^N$, $\boldsymbol{b}_{stim} \in \mathbb{R}^N$ correspond to baseline activity and background input. We used RNN models with 4, 8, and 12 hidden units for tasks **a, b, c**, respectively. Training details are provided in Appendix A.4.1. For each task, we applied the trained RNN model to perform 1000 trials, and recorded its hidden units activity across timesteps on every trial. Then we trained NetFormer to predict the next-step hidden units activity $\boldsymbol{h}_{k+1}$ based on the present and past hidden activity $(\boldsymbol{h}_{k-H}, \ldots, \boldsymbol{h}_k)$ and stimulus inputs $(\boldsymbol{s}_{k+1-H}, \ldots, \boldsymbol{s}_{k+1})$ on 800 of those trials, and held out the remaining 200 trials for evaluation. Compared to the linear

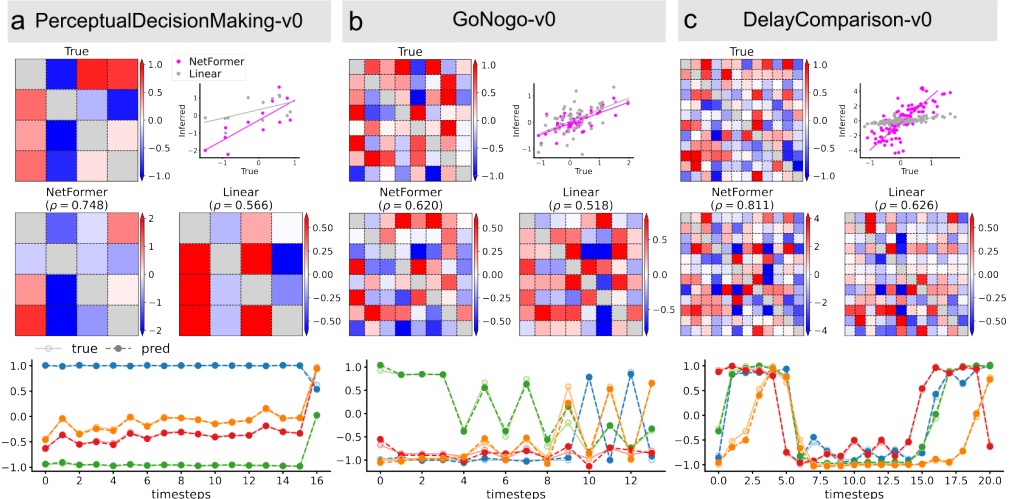

**Figure 4: NetFormer captures task-induced connectivity patterns. a. Top**: True connectivity matrix ($\boldsymbol{W}_{rec}$) from RNN trained to perform the Perceptual Decision Making task, followed by inferred connectivity from NetFormer's linearized attention matrix or linear regression weight matrix. For NetFormer, the inferred connectivity shown is the one whose Spearman correlation $\rho$ is closest to the average $\rho$ across 5 random seeds. Colorbars: scale of off-diagonal entries. Diagonal entries are masked in grey. **Inset**: Scatter plot of off-diagonal entries in the visualized inferred versus true connectivity matrix. **Bottom**: True (solid line, open circles) and NetFormer-predicted (dashed line, filled circles) activity for 4 example hidden units on a held-out trial. Predicted traces were obtained by concatenating one-step-ahead predictions. Hidden units are distinguished by colors. **b-c.** Same as **a**, for RNNs trained to perform the Go-Nogo task and Delay Comparision task, respectively.

regression model, NetFormer attains higher accuracy in both dynamics prediction and connectivity recovery (Figure 4 and Appendix A.4.2).

# 4 CONNECTIVITY-CONSTRAINED SIMULATION AND NEURAL DATA

Neurons form synapses based, in part, on factors controlled by their genetic and morphological *cell types*. The transmission of information through these synapses is influenced by activity history and behavioral states. To test its ability to handle these and allied complexities, we applied the NetFormer to a recent multi-modal dataset from (Bugeon et al., 2022), containing both simultaneously recorded neuronal activity and cell type information in the mouse primary visual cortex. After training the NetFormer to predict the neural activity, we used the time-averaged attention matrix as an inferred connectivity strength between neurons. We compared this inferred connectivity against an independent experimental measurement that serves as a cell-type level ground truth. This is the cell-type averaged postsynaptic potential (PSP) measured directly using paired patch-clamp experiments (Campagnola et al., 2022), in which the postsynaptic voltage responses of individual "downstream" neurons are recorded in response to spikes elicited in specific "paired" neurons. As an additional test, we observed that the inferred dynamical connectivity is more similar within the same behavioral state and more distinct between different states. Furthermore, following the measured cell-type level connectivity, we developed a connectivity-constrained simulation to produce neural dynamics with a fully specified ground truth connectivity. On this simulated dataset, we assessed the NetFormer's ability to infer both individual-neuron level and cell-type level connectivities. More details are in Appendix A.5.

## 4.1 DATASETS

**Multi-modal in-vivo neural recording.** The publicly available dataset (Bugeon et al., 2022) includes spontaneous population activity recorded from the mouse primary visual cortex (V1) via two-photon calcium imaging. We trained NetFormer models on data from one subject (SB025), which includes recordings of 2481 neurons. The dataset also provides single-cell spatial transcriptomics data, enabling identification of excitatory (EC) and three inhibitory neuron subclasses (Pvalb, Sst, Vip).

**Connectivity-constrained simulation.** We generated activity of a synthetic neuron population with 200 neurons whose cell-type level connectivity is constrained by the aforementioned patch clamp experiments (Campagnola et al., 2022). Specifically, we simulated the leaky-integration system

$$\frac{d}{dt}\boldsymbol{x}(t) = -\boldsymbol{x}(t) + \tanh\left(\boldsymbol{W}\boldsymbol{x}(t) + \boldsymbol{b}\right) + \boldsymbol{\epsilon} \tag{8}$$

using the forward Euler scheme with a step size $\delta = 1$, obtaining in discrete timesteps

$$\boldsymbol{x}_{k+1} = \tanh(\boldsymbol{W}\boldsymbol{x}_k + \boldsymbol{b}) + \boldsymbol{\epsilon}. \tag{9}$$

Here $\boldsymbol{\epsilon}$ stands for Gaussian observation noise, $\boldsymbol{b}$ represents the baseline activity, and $\boldsymbol{W}$ denotes the neuronal connectivity. In our simulation, $76\%$ of neurons are excitatory, with the remaining $24\%$ being inhibitory. The inhibitory neurons are further evenly subdivided into three cell types (Pvalb, Sst, and Vip). We used cell-type specific means and variances of PSPs measured in the patch clamp experiments to define Gaussian distributions of connection strengths between each pair of cell types, and sampled the connection strength between individual cells accordingly. We also provide an additional simulation in Appendix A.7, where the tanh is replaced with a sigmoid nonlinearity.

## 4.2 BASELINES AND EVALUATION METRICS

We benchmarked NetFormer against multiple baselines: (i) a linear recurrent model (referred to as "linear regression"), where $\boldsymbol{x}_{k+1} = \boldsymbol{W}\boldsymbol{x}_k + \boldsymbol{b}$; two variants of nonlinear recurrent neural networks (referred to as "RNNs"): (ii) $\boldsymbol{x}_{k+1} = \tanh(\boldsymbol{W}\boldsymbol{x}_k + \boldsymbol{b})$, which matches the connectivity-constrained simulation and serves as an "oracle" type model giving a corresponding upper bound on performance, (iii) $\boldsymbol{x}_{k+1} = \exp(\boldsymbol{W}\boldsymbol{x}_k + \boldsymbol{b})$, of the form of commonly used generalized linear models (GLMs) in neuroscience (Pillow et al., 2008), with a nonlinearity mismatched to that of the simulation itself. We also considered standard statistical metrics, including cross-correlation, covariance, mutual information, and transfer entropy (with details in Appendix A.6). We evaluated activity prediction using mean squared error (MSE), coefficient of determination ($R^2$), and the Pearson correlation coefficient. We assessed the correlation between inferred and true connectivity using Pearson and Spearman correlation coefficients, both at the $N \times N$ neuron level ($N$: number of recorded neurons) and the $K \times K$ cell type-level ($K = 4$ includes one excitatory and three inhibitory types: Pvalb, Sst, Vip). More details are provided in Appendix A.10.2.

## 4.3 TIME-AVERAGED CONNECTIVITY INFERENCE

**NetFormer outperforms baselines in inferring time-averaged connectivity.** As shown in Table 1 and Figure 5a, on both simulated and real neuronal activity data, NetFormer outperforms other baseline models in predicting activity and inferring connectivity in the majority of the evaluations. We hypothesize that NetFormer's advantage relative to other baseline methods mainly stem from two key aspects. The first is that the underlying connectivity in the NetFormer model is nonstationary rather than static, as assumed by the other baseline models and methods. The second is the absence of a need to specify an activation nonlinearity. Notable, in the connectivity-constrained simulation, NetFormer achieves comparable performance to the "oracle" model (RNN with tanh nonlinearity). By contrast, the RNN with an exponential nonlinearity, misspecified with respect to the simulation, performed significantly worse. This contrast illustrates a strength of the NetFormer model: NetFormer does not involve specification of an activation function, avoiding the fragility that this can entail. We further tested NetFormer's robustness to spurious correlations following Das & Fiete (2020) (see details in Appendix A.9). NetFormer shows some resilience to spurious correlations, but would also struggle when the recurrent weights are too strong or too weak, similar to other methods studied in Das & Fiete (2020), including the GLM (Pillow et al., 2008), logistic regression (Lee et al., 2006), and the Ising model (Roudi et al., 2009).

**Excitation cell type can be decoded from learned positional embeddings.** Following Mi et al. (2024), we also observe that the learned positional embeddings $\boldsymbol{E}$ in the NetFormer model can be used to decode an aspect of cell class information. Specifically, using learned embeddings of neurons in the training set, we trained a binary classifier via logistic regression to classify neurons as excitatory or inhibitory, a coarser grouping which subsumes the genetic/morpological categories considered above. When applied to classify held-out neurons on the test set, the trained classifier attains $100\%$ top-1 accuracy in simulation data, $66.67\%$ top-1 accuracy and AUROC score of $0.700$ in real data (see confusion matrices in Figure 5b). This shows that the learned positional embeddings are linearly separable according to this aspect of neurons' cell type identity.

**NetFormer demonstrates robustness against partial observation.** For most biological neuronal circuits, each recording can only access the activity of a subset of all neurons. To evaluate its robustness against such partial observation, we fitted NetFormer to a randomly selected subset of neurons in the simulated dataset. In Figure 5c, we show that the performance in recovering the

| | | | NetFormer | linear regression | RNN w/ tanh | RNN w/ exp | cross correlation | covariance | mutual information | transfer entropy |
|---|---|---|---|---|---|---|---|---|---|---|
| **simulation** | connectivity N × N | Pearson | **0.869**±0.002 | 0.817±0.002 | 0.905±0.000* | 0.581±0.011 | 0.823 | -0.029 | 0.539 | 0.600 |
| | | Spearman | **0.532**±0.001 | 0.507±0.001 | 0.546±0.000* | 0.393±0.009 | 0.519 | -0.015 | 0.262 | 0.339 |
| | connectivity K × K | Pearson | 0.879±0.001 | 0.885±0.001 | 0.908±0.000* | 0.887±0.008 | **0.888** | -0.438 | 0.371 | 0.419 |
| | | Spearman | **0.860**±0.002 | 0.852±0.005 | 0.866±0.002* | 0.822±0.025 | 0.732 | -0.334 | 0.018 | 0.353 |
| **in-vivo recording** | connectivity K × K | Pearson | **0.777**±0.047 | -0.395±0.020 | -0.395±0.036 | -0.407±0.006 | -0.017 | -0.162 | -0.176 | 0.075 |
| | | Spearman | **0.847**±0.063 | -0.409±0.051 | -0.343±0.105 | -0.191±0.300 | -0.080 | -0.190 | -0.061 | 0.233 |
| | activity prediction | MSE | **0.404**±0.004 | 0.443±0.001 | 0.560±0.001 | 0.476±0.003 | – | – | – | – |
| | | Pearson | **0.740**±0.003 | 0.720±0.001 | 0.639±0.000 | 0.699±0.002 | – | – | – | – |
| | | $R^2$ | **0.548**±0.004 | 0.515±0.001 | 0.386±0.001 | 0.478±0.004 | – | – | – | – |

**Table 1: NetFormer outperforms classical baselines methods in dynamics prediction and connectivity inference.** Results from both connectivity-constrained simulation and neural recording. An asterisk (*) indicates that the RNN with tanh activation serves as the oracle model (upper bound for performance on the simulation data). Results from mutual information and transfer entropy are compared against the absolute values of ground truth connectivity. Simulation data has ground truth for both neuron-level ($N \times N$) and cell type-level ($K \times K$) connectivity. Patch-clamp results serve as the ground truth for real neural data cell-type level connectivity. Connectivity inference is assessed with Spearman's and Pearson's correlations. Next-step activity prediction is evaluated with MSE, Pearson's coefficient, and $R^2$ on the test set.

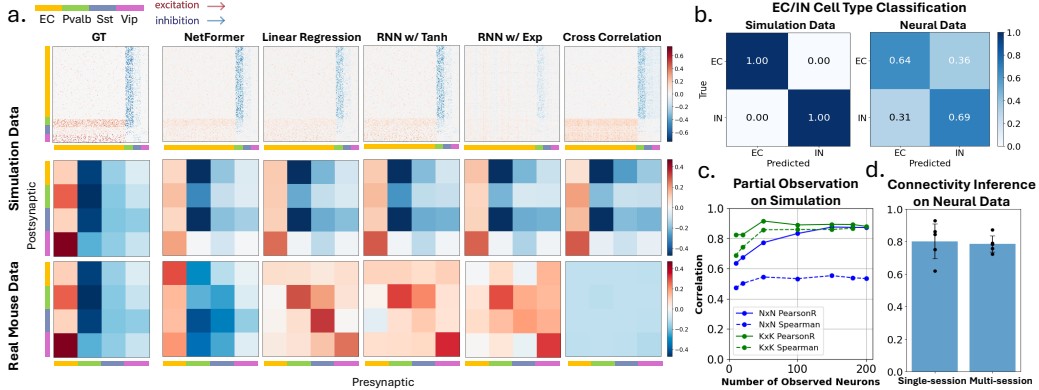

**Figure 5: a.** Visualization of true and inferred connectivities at both neuron level and cell-type level in simulation data and neural data. NetFormer is benchmarked with linear regression, RNNs, and standard statistical metrics. A positive linear transformation has been applied to standardize all matrices to the same range for better visualization. **b.** Confusion matrices of excitatory (EC) and inhibitory (IN) cell type classification on both connectivity-constrained simulation and neural data. **c.** Experiment on different levels of partial observability in the connectivity-constrained simulation. Connectivity inference is evaluated at both neuron level and cell-type level. **d.** Connectivity inference in real neural data with NetFormer models trained on a single session and multiple sessions from the same subject. Errorbars: Spearman correlations across 5 random seeds.

neuron-level connectivity does not decrease significantly even with only half of the neurons observed. Cell-type level connectivity inference is more robust against such partial observations, highlighting the potential of NetFormer to effectively derive cell-type level connectivity from real neural data.

**NetFormer can fit neural recordings across sessions.** RNN-type models, designed to recover connectivity at the level of individual neurons, cannot easily incorporate data of varying population sizes across experimental sessions. In contrast, NetFormer promotes scalability by allowing parameter sharing ($\boldsymbol{W}_Q$, $\boldsymbol{W}_K$) across sessions. These parameters are defined in the temporal dimension and thus do not increase with the number of neurons. Figure 5d shows that NetFormer achieves comparable performance for connectivity inference when both fitting a single session and multiple sessions.

## 4.4 NONSTATIONARY CONNECTIVITY INFERENCE

Building on NetFormer's ability to capture nonstationary connectivity changes over time in Section 3, we extended this evaluation to real neural data. In this dataset (Bugeon et al., 2022), activity at each timestep is labeled by one of the three behavioral states: running, stationary desynchronized, and stationary synchronized. Figure 6a shows that neurons of different cell types can exhibit different activity patterns across these states, suggesting that neural activity is informative of behavioral states. Thus informed, we explored whether the time-varying attention could capture connectivity changes among these states (see details and visualizations in Appendix A.8). We compared NetFormer with two baseline methods capable of capturing nonstationary connectivity: a low-tensor-rank RNN (LtrRNN) (Pellegrino et al., 2023) and an autoregressive Hidden Markov Model (AR-HMM) (Fox et al., 2008; Linderman et al., 2017b). LtrRNN model the connectivity at each "trial" as a combination of rank-1 matrices. Since our neural data is measured during spontaneous activity without an explicit trial structure, to apply LtrRNN, we constructed "trials" using sliding windows on the data. Unlike

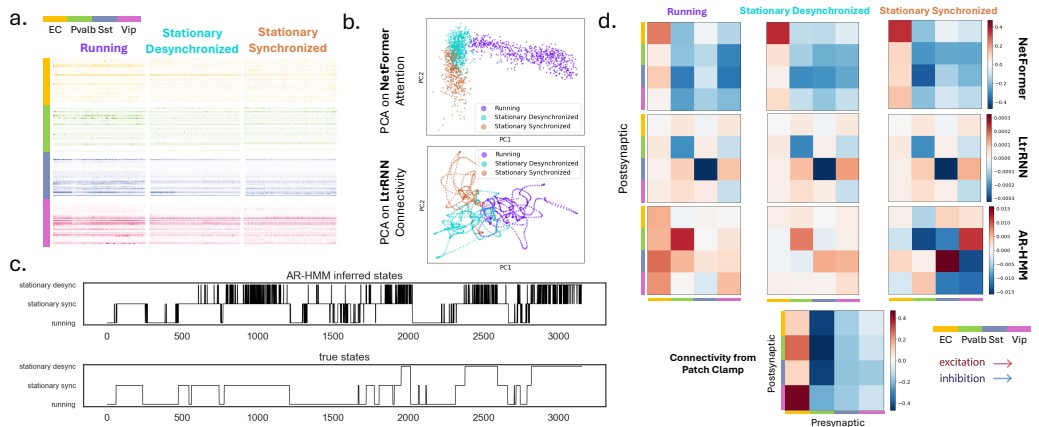

**Figure 6: Connectivity inference across behavioral states. a.** Neural activities across cell types in three behavioral states. **b.** Projection of NetFormer attention maps and LtrRNN-inferred connectivity weights onto top two principal components. Each dot represents a different timestep, colored by behavioral states. PCA on NetFormer attention maps shows greater similarity between the two stationary states compared to the running state. **c.** Comparison of AR-HMM inferred states with true behavioral states. **d.** Comparison of inferred state-specific connectivities across NetFormer, LtrRNN, and AR-HMM. NetFormer-inferred connectivities are in better agreement with the patch-clamp result.

NetFormer which shares parameters across time/trials, the number of parameters LtrRNN has to learn scales linearly with the number of trials. As a second baseline, we considered the AR-HMM, which assumes that there are discrete latent states switching underlying the observed activity, and each state admits unique dynamics through a different connectivity matrix. Due to its discrete state switching mechanism, it is not suitable for capturing continuous connectivity changes, and its state discovery relies heavily on the user-specified number of states.

While NetFormer-inferred cell-type level connectivity is in good agreement with the patch-clamp experimental ground truth, connectivities inferred by LtrRNN and AR-HMM bear little correlation to this ground truth (Figure 6d). A quantitative comparison is provided in Table 6, Appendix A.8. Notably, consistent with prior experimental observations (Fu et al., 2014), attention also witnesses an increase in inhibitory activity from presynaptic Vip neurons and a decrease in inhibition from presynaptic Sst neurons during the running state, as seen by the darker Vip column in the running state compared to the stationary states and the lighter Sst column (Figure 6d, top row). Figure 6b further demonstrates that state information is implicitly captured by NetFormer's attention maps, showing greater similarity within states and clear distinctions between states. Notably, stationary desynchronized and stationary synchronized states show more similarity to each other than to the running state. Moreover, compared to weights inferred by LtrRNN, PCA on NetFormer-inferred weights yields a cleaner separation between running and two stationary states. When tasked to find three states from the neural data, those inferred by the AR-HMM largely align with the three behavioral states, albeit with higher noise (Figure 6c).

## 5 CONCLUSION AND DISCUSSION

Experience, activity, and adaptation change the effective connectivity of biological neuronal networks via mechanisms including synaptic plasticity and neuromodulation, all playing out across various timescales. This perspective poses connectivity as a dynamical variable that should be tracked, rather than inferred once. Here, we propose the NetFormer as a light-weight model for dynamical connectivity inference. We began with a mathematical analysis that relates nonlinear and nonstationary dynamics to its linearized attention mechanism. We further demonstrated, on representative simulated and in-vivo neural datasets, the strength of our model through comparison against various baselines to predict nonlinear neural dynamics and to capture the underlying dynamical connectivity.

This said, our method has several limitations: (i) Partial observability of neuronal population dynamics has been a major confounding factor for connectivity inference, and our method is no exception. (ii) As our model learns the forward dynamics through a history-dependent linearization of the system in a local temporal neighborhood, its ability to capture nonlinear or nonstationary systems is limited compared to fully nonlinear or layered transformer-type models. Despite these limitations, our work presents a step forward to addressing the long-standing challenge of extracting nonstationary neuronal network structure from complex functional data, and brings new insights into the interpretability of the transformer model and its applicability in modeling nonstationary dynamical systems.

## ACKNOWLEDGMENTS

This work was supported by the NSF grant NCS-FO #2024364 and the NIH grant 1RF1DA055669 (ZL and ESB) and the Shanahan Foundation Fellowship (WZ and LM). TL acknowledges the support in part by A3D3 NSF grant OAC-2117997 and the Department of Electrical and Computer Engineering at the University of Washington. HW is partially supported by Amazon Faculty Research Award, Microsoft AI & Society Fellowship, NSF CAREER Award IIS-2340125, NIH grant R01CA297832, and NSF grant IIS-2127918.

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

# A APPENDIX

## A.1 JUSTIFICATION FOR LINEARIZED ATTENTION APPLIED TO "LEAKY" SYSTEMS (SEC 2, EQN 6)

Using the forward Euler method and step size $\delta$, Equation 6 can be simulated as

$$\boldsymbol{x}_{k+1} = \boldsymbol{x}_k + \delta\Big( -\boldsymbol{x}_k + f(\boldsymbol{W}_k\boldsymbol{x}_k)\Big) = \boldsymbol{x}_k - \delta\boldsymbol{x}_k + \delta f(\boldsymbol{W}_k\boldsymbol{x}_k). \quad (10)$$

Following the same sigmoidal assumption on $f$, Equation 10 can be written as

$$\boldsymbol{x}_{k+1} = \boldsymbol{x}_k + \delta(f'(0)\boldsymbol{W}_k - \boldsymbol{I})\boldsymbol{x}_k + \delta O(\boldsymbol{x}_k^3), \quad (11)$$

where $\boldsymbol{I}$ is the $N \times N$ identity matrix. Therefore, the linearized attention matrix $\boldsymbol{A}_k$ learned from Equation 3 may reflect the true interactions $\boldsymbol{W}_k$ by approximating $\delta(f'(0)\boldsymbol{W}_k - \boldsymbol{I})$, and can capture the interactions between different variables (off-diagonal entries of $\boldsymbol{W}_k$) up to a scaling factor $(\delta f'(0))$.

## A.2 ADDITIONAL DETAILS FOR NONLINEAR AND NONSTATIONARY SYSTEMS SIMULATION (SEC 3.1)

### A.2.1 SIMULATION DETAILS

In Figure 2 **a, b**, ground-truth $\boldsymbol{W}$ were generated randomly, with real-part of each eigenvalue clipped at 0 to ensure stability of the system. $\boldsymbol{W}$ in **a, b** were also used as $\boldsymbol{W}_0$ in **c,d**, respectively. $\boldsymbol{\omega}$ in **c, d** were picked randomly while maintaining stability of the system. In **a**, the system trajectory was simulated using the closed-form solution $\boldsymbol{x}(t) = e^{\boldsymbol{W}t}\boldsymbol{\eta}$, where $\boldsymbol{\eta}$ is the initial state. In **b-d**, trajectories were simulated using the forward Euler method: $\boldsymbol{x}_{k+1} = \boldsymbol{x}_k + \delta f(\boldsymbol{W}_k\boldsymbol{x}_k)$, where $\boldsymbol{W}_k \equiv \boldsymbol{W}$ for **b**, and $\boldsymbol{W}_k = \boldsymbol{W}_0 + \boldsymbol{x}_k\boldsymbol{\omega}^\top$ for **c, d**. All simulations consist of 3000 timesteps with stepsize $\delta = 0.01$, with the first 80% used as training set, and last 20% as test set. For **c, d**, ground-truth connectivity matrices were computed as the time-averaged connectivity across test set timesteps $\bar{\boldsymbol{W}} = \sum_{k=2400}^{3000} \boldsymbol{W}_k$ . Simulated trajectories are visualized in Figure 7. In all settings, the NetFormer model was trained to minimize the mean squared error (MSE) on the trainining set for 1100 epochs using the Adam optimizer in Pytorch, with $H = 1, M = 5, D = 5$, batch size = 80, initial learning rate = 0.01. In **b, d**, learning rate was decayed by a factor of 0.9 every 100 epochs. In **c**, learning rate was decayed by a factor of 0.8 every 100 epochs.

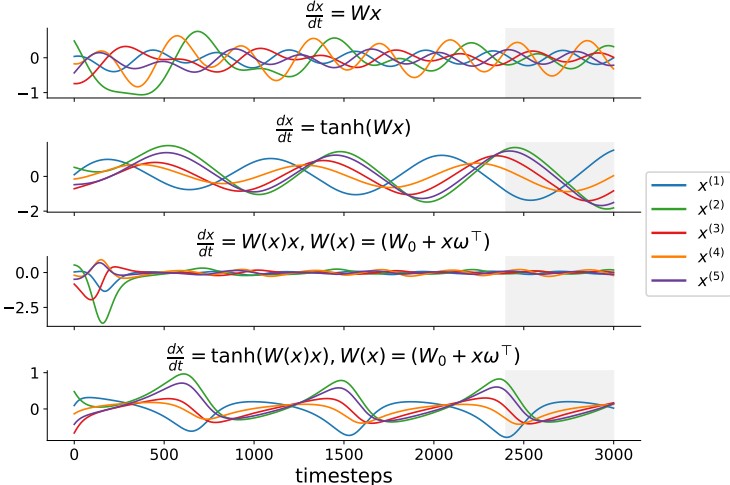

**Figure 7:** Simulated trajectories of toy models in Figure 2. Shaded regions represent timesteps used as test set.

### A.2.2 QUANTITIVE COMPARSION WITH LINEAR REGRESSOIN MODEL

For each toy system in section 3.1, we trained 10 NetFormer models with different random seeds (initializations), and computed the Spearman's rank correlation coefficient ($\rho$) and the Pearson correlation coefficient ($r$) between the off-diagonal entries of $\bar{A}$ and $\bar{W}$ for each trained model. In terms of $\rho$, $\bar{A}$ achieved comparable performance as $A_{\mathrm{OLS}}$ in systems **a, b** ($p > 0.3$, two-sided one sample t test), but significantly outperformed $A_{\mathrm{OLS}}$ in systems **c, d** ($p < 10^{-8}$) (Figure 8 left). Similar observations can be made with $r$ (Figure 8 right). For each system, the linearized attention matrix visualized in Figure 2 is the one whose $\rho$ is the closet to the average $\rho$ across 10 random seeds.

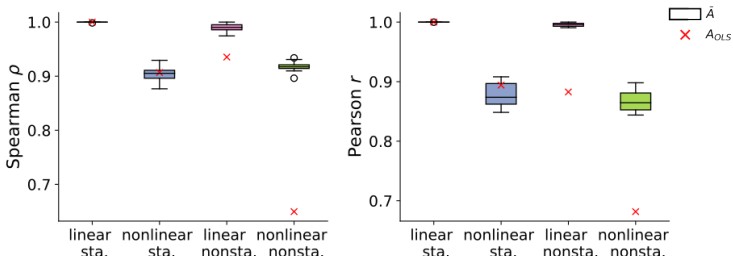

**Figure 8:** Comparison between $A_{\mathrm{OLS}}$ (red cross) and $\bar{A}$ from NetFormer models with 10 different random initializations (boxplots).

### A.2.3 NONSTATIONARITY CONNECTIVITY TRACKING

On toy systems with nonstationary connectivity (Figure 2c, d), we evaluated how well linearized attention matrices across timesteps can track changes in the connectivity. For each pair $(i, j)$, $i, j = 1, \ldots, 5, i \neq j$, we collected $A_{ij}$ and $W_{ij}$ across all test timesteps, resulting in two time-varying series $A_{ij}(t)$ and $W_{ij}(t)$, and computed the Pearson correlation coefficient between them. Results for 10 trained NetFormer models with different random seeds are shown in Figure 9. Distributions of the temporal correlation coefficients for all off-diagonal pairs $(i, j)$ are shown as violin plots, where each violin corresponds to model trained with one random seed. The median of each distribution is marked with a black line. All medians are greater than 0.999.

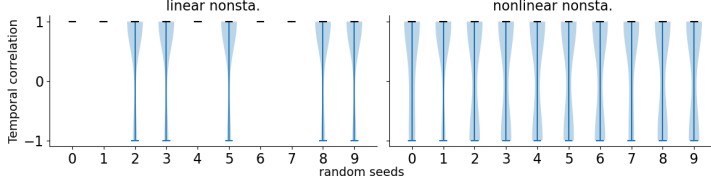

**Figure 9:** Distribution of test set temporal correlation between the linearized attention matrix and the true nonstationary connectivity. Each column shows result from NetFormer model with a different random seed. Median of each distribution is marked in black. All medians are greater than 0.999.

### A.2.4 FURTHER DISCUSSION ON NONLINEAR DYNAMICAL SYSTEMS

In Section 2, we showed that when $f$ is sigmoidal, $A$ can reflect $W$ through Taylor series approximation of $f(W x_k)$. Take $f = \tanh$ as an example. Let $w_i$ denote the $i$-th row of $W$. When $|w_i^\top x_k| < \frac{\pi}{2} \ \forall i = 1, \ldots, N$,

$$\tanh(W x_k) = \tanh(0) + \tanh'(0) W x_k + O(x_k^3).$$

As $\tanh(0) = 0$, the forward Euler method is

$$x_{k+1} = x_k + \delta \tanh(W x_k) = x_k + \delta \tanh'(0) W x_k + \delta O(x_k^3).$$

This analysis also applies to other sigmoidal functions $f$, such as arctan, with

$$f(0) = 0, f(\xi) = f'(0)\xi + O(\xi^3) \text{ for } \xi \text{ within some interval around } 0.$$

Therefore, we hypothesize that $A$ can capture $W$ through learning $\delta f'(0)W$ for sigmoidal $f$. It is also clear that learning $W$ becomes more challenging when $w_i^\top x$ does not always stay within the radius of convergence of the Maclaurin series. Nonetheless, we note that if some $w_i^\top x$ is constantly outside the convergence region, $f(w_i^\top x)$ will be constantly positive or negative, and the system will either blow up or decay to zero. Therefore, for the systems of interest here, which are those with interesting persistent dynamics, from time to time $w_i^\top x$ must fall within the convergence region where the Maclaurin series representation is valid. That being said, while $A$ may still be able to capture some aspect of $W$, it could become less accurate, and may require more observations of the system to gather sufficient timesteps within the convergence region.

In the example nonlinear dynamical system shown in Figure 2b, $w_i^\top x_k$ stays within the radius of convergence of $\tanh$ for all $i$ and $k$, which makes the Maclaurin series approximation valid for all timesteps. In Figure 10, we provide another toy model example showing that the attention from NetFormer still bears considerable similarity to the ground-truth $W$ even when $w_i^\top x_k$ falls out of the convergence region for some $i$ and $k$.

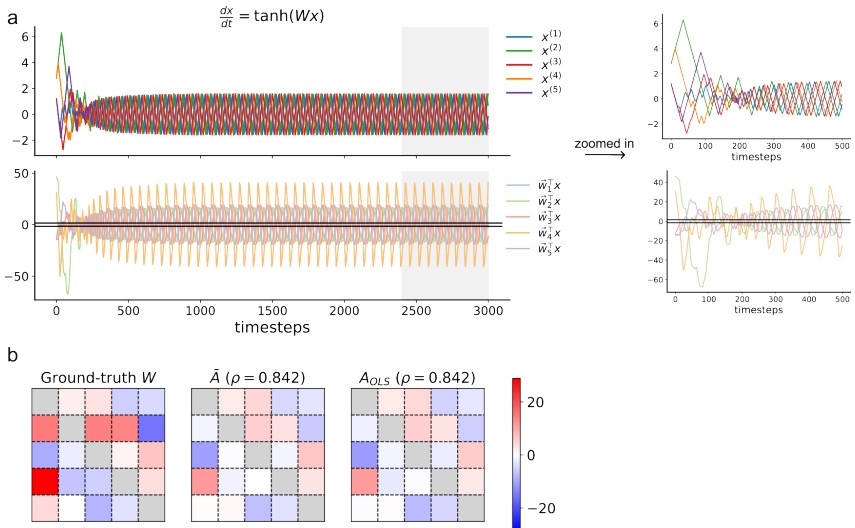

**Figure 10:** Demonstration of NetFormer on a nonlinear system $\frac{d\boldsymbol{x}}{dt} = \tanh(\boldsymbol{W}\boldsymbol{x})$ where $w_i^\top x$ does not always stay within the convergence region of the Maclaurin series of $\tanh$. **a. Top row:** Trajectories of the simulated system. Simulation was done using the forward Euler method with stepsize $\delta = 0.1$. Shaded regions represent timesteps used as test set. **Bottom row:** Visualization of $w_i^\top x$ across simulated timesteps. Boundaries of the convergence region, $\pm\frac{\pi}{2}$, were marked with black horizontal lines. The right column provides a zoomed-in view of the first 500 simulation timesteps. **b. Left to right** Ground-truth $\boldsymbol{W}$, average linearized attention matrix across test timesteps from NetFormer ($\bar{\boldsymbol{A}}$), $\boldsymbol{A}_{\text{OLS}}$ fitted through least-squares regression. Colorbars indicate the scale of the off-diagonal entries, and the diagonal entries are masked in grey. $\bar{\boldsymbol{A}}$, $\boldsymbol{A}_{\text{OLS}}$ were rescaled for visualization. Spearman's rank correlation coefficients ($\rho$) were computed between the off-diagonal entries of $\bar{\boldsymbol{A}}$ or $\boldsymbol{A}_{\text{OLS}}$ and $\boldsymbol{W}$. We trained 10 NetFormer models with different random initializations ($\rho = 0.841 \pm 0.02$, mean $\pm$ std), and $\bar{\boldsymbol{A}}$ shown is the one whose $\rho$ is the closet to the average $\rho$ across 10 random initializations. NetFormer models achieved similar performance as $\boldsymbol{A}_{\text{OLS}}$ ($p = 0.9$, two-sided one sample t test). All NetFormer models were trained to minimize the mean squared error on the trainining set for 600 epochs using the Adam optimizer in Pytorch, with $H = 1, M = 5, D = 5$, batch size = 80. Learning rate was initialized to 0.01, and was decayed by a factor of 0.9 every 100 epochs.

A.3    ADDITIONAL DETAILS FOR STDP SIMULATION (SEC 3.2)

Our STDP simulation is largely based on Neuromatch Academy (2023). The dynamics of the LIF neuron follows

$$\tau_m \frac{dV}{dt} = -(V - E_L) - g_E(t)(V - E_E) \tag{12}$$

$$V(t) \geq V_{th} \Rightarrow V(t) = V_{reset} \tag{13}$$

where $V$ is the membrane potential, $\tau_m$ is the membrane time constant, $E_L$ is the resting potential, $E_E$ is the synapse reversal potential, $V_{th}$ is the spiking threshold, and $V_{reset}$ is the reset potential. Once $V(t)$ crosses the spiking threshold, we say the LIF neuron emits a spike, and $V(t)$ will be reset to and held at $V_{reset}$ for a refractory period $t_{ref}$. $g_E(t)$ is the total excitatory synaptic conductance, which is the total conductance of all active pre-synapses (i.e. synapses coming into the LIF neuron) at that time:

$$g_E(t) = \sum_{i=1}^{N} g_i(t)\delta_i(t - t_{spk}) \tag{14}$$

where $\delta_i$ is the delta function: $\delta_i(t - t_{spk}) = 1$ if there is a spike at pre-synapse $i$ at time $t$, and 0 otherwise. The conductance of each pre-synapse $g_i$ evolves following

$$\frac{dg_i}{dt} = -\frac{g_i}{\tau_E} + \bar{g}_i\delta_i(t - t_{spk}) \tag{15}$$

where $\tau_E$ is the EPSP time constant, and $\bar{g}_i$ is the peak synaptic conductance of pre-synapse $i$. All $\bar{g}_i$ are bounded between 0 and $\bar{g}_{max}$.

When there is a spike arriving at the $i$-th pre-synapse,

$$\bar{g}_i = \bar{g}_i + M(t)\bar{g}_{max} \tag{16}$$

where $M(t)$ helps tracking the time since the last postsynaptic spike emitted by the LIF neuron. When the postsynaptic LIF neuron spikes, all pre-synapses are updated:

$$\bar{g}_i = \bar{g}_i + P_i(t)\bar{g}_{max}, \forall i \tag{17}$$

where $P_i(t)$ helps tracking the time since the last spike at the $i$-th pre-synapse. STDP is enforced through $M(t)$ and $P_i(t)$. Specifically, $M(t)$ follows

$$\tau_- \frac{dM}{dt} = -M \tag{18}$$

and whenever the postsynaptic LIF neuron spikes,

$$M(t) = M(t) - A_-. \tag{19}$$

$P_i(t)$ follows

$$\tau_+ \frac{dP}{dt} = -P \tag{20}$$

and whenever the $i$-th presynaptic neuron spikes,

$$P(t) = P(t) + A_+. \tag{21}$$

$\tau_+$ and $\tau_-$ specify the range of separation between pre- and postsynaptic spikes where STDP takes effect. $A_+, A_-$ are both positive, and define the maximum amount of synaptic strengthening and weakening, respectively. It follows that $M(t) \leq 0, P_i(t) \geq 0 \ \forall t$. $M(t)$ and $P_i(t)$ effectively capture the STDP rule

$$\Delta W = A_+ e^{(t_{pre} - t_{post})/\tau_+} \text{ if } t_{post} > t_{pre} \tag{22}$$

$$\Delta W = -A_- e^{-(t_{pre} - t_{post})/\tau_-} \text{ if } t_{post} < t_{pre} \tag{23}$$

as shown in Figure 3a. The constant parameters and initial conditions in our simulation are set in the same way as in Neuromatch Academy (2023), and are summarized in tables 2, 3. Pre-synaptic spike trains are modeled as independent Poisson processes with rate 50Hz.

To generate data, we run the simulation for 100,000 timesteps where each timestep corresponds to 1ms. We used the first 80% of data for training, and the last 20% of data as the test set. Membrane potential of the LIF neuron was then z-scored using its mean and std on the training set. We fitted 5 NetFormer models using different random seeds. All NetFormer models have $H = 5, M = 101, D = 101$, and were trained to minimized the mean squared error on the training set for 20 epochs using the Adam optimizer with learning rate 0.005, batch size 64 in Pytorch. Result from one seed is visualized in Figure 3d-f. We used a sliding window of length 10,000 timesteps to smooth both true and inferred synaptic weight trajectories before computing the correlation coefficients.

**Table 2:** Constant parameters in STDP simulation

| Parameter | Value |
| --- | --- |
| $\tau_m$ | 10 [ms] |
| $E_L$ | -75 [mV] |
| $E_E$ | 0 [mV] |
| $V_{th}$ | -55 [mV] |
| $V_{reset}$ | -75 [mV] |
| $t_{ref}$ | 2 [ms] |
| $\tau_E$ | 5 [ms] |
| $\bar{g}_{max}$ | 0.024 |
| $\tau_+$ | 20 [ms] |
| $\tau_-$ | 20 [ms] |
| $A_+$ | 0.008 |
| $A_-$ | 0.0088 |

**Table 3:** Initial conditions in STDP simulation

| Variable | Initial value |
| --- | --- |
| $V$ | -65 [mV] |
| $g_i$ | 0.014 |
| $M$ | 0 |
| $P$ | 0 |

## A.4 ADDITIONAL DETAILS FOR TASK-DRIVEN POPULATION ACTIVITY SIMULATION (SEC 3.3)

### A.4.1 SIMULATION DETAILS

The hidden dynamics of RNN models follows equation 7, and at each timestep $k$, the network activity is read out through a linear mapping

$$\boldsymbol{y}_k = \boldsymbol{W}_{out}\boldsymbol{h}_k. \tag{24}$$

All RNN models are trained to minimize the cross entropy loss using the Adam optimizer in Pytorch. In the Perceptual Decision Making task, we trained a RNN with 4 hidden units for 5000 epochs using learning 0.005, batch size 32. In the Go-Nogo task, we trained a RNN with 8 hidden units for 2000 epochs using learning 0.01, batch size 32. In the Delay Comparision task, we trained a RNN with 12 hidden units for 5000 epochs using learning 0.01, batch size 32. All trained RNNs achieve over 90% accuracy in the 1000 test trials. All trials are generated using the default parameters in the NeuroGym toolkit (Molano-Mazon et al., 2022).

In each task, we recorded the hidden units activity of the trained RNN during the 1000 test trials. We then trained NetFormer to predict the next-step hidden units activity based on the present and past $H$-step hidden activity and stimulus inputs. Hidden units activity during 800 trials were used for training, and the remaining 200 trials were using for evaluation. In each task, we trained 5 NetFormer models from different initializations to minimize the mean squared error (MSE) on the training set using the Adam optimizer in Pytorch. In the Perceptual Decision Making task, NetFormer has $H = 5$, $N = M = D = 7$, and was trained using learning rate 0.0025, batch size 64 for 100 epochs. In the Go-Nogo task, NetFormer has $H = 1$, $N = M = D = 11$, and was trained using learning rate 0.01, batch size 64 for 50 epochs. In the Delay Comparision task, NetFormer has $H = 5$, $N = M = D = 14$, and was trained using learning rate 0.005, batch size 64 for 50 epochs.

### A.4.2 QUANTITIVE COMPARSION WITH LINEAR REGRESSOIN MODEL

Table 4 provides a quantatitive comparison between NetFormer and the linear regression model.

| | Dynamics prediction | | | | Connectivity recovery | | | |
|---|---|---|---|---|---|---|---|---|
| | NetFormer | | Linear | | NetFormer | | Linear | |
| | MSE | $R^2$ | MSE | $R^2$ | Spearman | Pearson | Spearman | Pearson |
| **a** | **0.000**±0.000 | **0.998**±0.001 | 0.014 | 0.920 | **0.694**±0.163 | **0.760**±0.120 | 0.566 | 0.548 |
| **b** | **0.010**±0.000 | **0.972**±0.001 | 0.013 | 0.960 | **0.622**±0.006 | **0.722**±0.004 | 0.518 | 0.553 |
| **c** | **0.001**±0.000 | **0.997**±0.001 | 0.034 | 0.897 | **0.811**±0.011 | 0.633±0.105 | 0.626 | **0.650** |

Table 4: **a, b, c** corresponds to the Perceptual Decision Making task, Go-Nogo task, and Delay Comparision task, respectively. MSE and $R^2$ were evaluated on concatenated held-out trials. Spearman and Pearson correlation coefficients were computed between the off-diagonal entries of the inferred and true connectivity matrices. NetFormer results are the mean±std across 5 random seeds.

## A.5   Connectivity-constrained simulation and neural data: Datasets and preprocessing (Sec 4.1)

### A.5.1   Connectivity-constrained simulation

To construct the ground-truth connectivity matrix, for each cell-type pair and for every pair of neurons, we first drew a random sample from the uniform distribution between 0 and 1. Then, we used the connectivity probability from patch-clamp experiments Campagnola et al. (2022) as a cutoff threshold to determine if two neurons are connected. For connected neurons, we sampled their connection strength from a normal distribuion $\mathcal{N}(\mu, 0.1)$, where $\mu$ is the measured post-synaptic potential from patch-clamp experiments. We simulated $30,000$ steps for 200 neurons, using the first $80\%$ timesteps for training and the last $20\%$ for testing.

### A.5.2   Patch-clamp dataset

The dataset released in Campagnola et al. (2022) contains experimental results of connectivity probability and connectivity strength (Postsynaptic Potential (PSP)) at the cell-type level measured using patch-clamp. In each experiment, up to eight neurons were simultaneously subjected to whole-cell patch-clamp recording, mainly under current-clamp conditions, with some stimuli also tested under voltage-clamp conditions. Stimuli were applied to each patched neuron while recording the other neurons for postsynaptic responses. We mainly focus on the experimental results for layers 2/3 in mouse primary visual cortex (V1), to match the neurons recorded in the multimodal mouse datatset (Bugeon et al., 2022).

### A.5.3   Multimodal neural recording

For functional activity recordings from neuronal populations, we used a recent, public multimodal dataset provided by Bugeon et al. (2022). This dataset includes spontaneous population activity recordings from the mouse primary visual cortex (V1) across layers 2/3 via 2-photon calcium imaging at a temporal sampling frequency of 4.3Hz across six 20-minute sessions, recording approximately $500$ neurons per session. Spatial coordinates of the recorded neurons are also provided. We trained our models on data from one experimental subject (SB025), which includes recordings of $2481$ neurons, with some neurons repeating across six sessions. The dataset also includes single-cell spatial transcriptomics, profiling mRNA expression for 72 selected genes to identify excitatory and inhibitory class labels of neurons. $51\%$ of neurons in the inhibotiry class can further be identified to be one of Lamp5, Pvalb, Vip, Sncg, and Sst.

### A.5.4   Data preprocessing

In the connectivity-constrained simulation, when using RNN with an exponential activation, we added the minimum activity value to all simulated neuronal activity to ensure nonnegativity.

For the experimental neural recording, which is nonnegative, we normalized it using the mean and standard deviation calculated across all sessions and neurons involved in training. When using the RNN model with exponential activation, we normalized the data by dividing by the standard deviation only, without first subtracting the mean.

## A.6 BASELINES

### A.6.1 STATIONARY CONNECTIVITY BASELINES (SEC 4.2)

**Linear Regression:** We denote neural activity data as $X_k = [x_{k-H+1} \quad \cdots \quad x_k] \in \mathbb{R}^{N \times H}$ recorded from $N$ neurons and $H$ time steps. Let $x_{k+1} \in \mathbb{R}^N$ denote the neuronal activity at the $(k+1)$th time step. Given previous 1 time step, $x_k$, linear regression predicts current time step activity as

$$\hat{x}_{k+1} = W x_k + b$$

**Recurrent Neural Network (RNN) with tanh activation:** Given neuronal activity across previous $p$ time steps, $x_k, x_{k-1}, \ldots, x_{k-p+1}$, a RNN with predefined Tanh activation function predicts current time step activity as

$$\hat{x}_{k+1} = \sigma \left( W^{(0)} x_k + W^{(1)} x_{k-1} + \cdots + W^{(p-1)} x_{k-p+1} + b \right), \sigma = \tanh$$

where $W^{(l)}, l \in \{0, 1, \ldots, p-1\}$, represent how the previous $l$-th step affects the current step activity and each element $W_{ij}^{(l)}$ represents how the $j$-th neuron at the previous $l$-th time step influences the $i$th neuron in current step. The RNN is trained by minimizing mean squared errors (MSE) of the current time step activity prediction given previous time steps. $p = 1$ is commonly used for RNN. For modeling both simulation data and real mouse data, we chose $p = 1$, because the simulation data has exactly one timestep dependency. Using larger $p$ did not improve performance in real data either.

**Recurrent Neural Network (RNN) with exponential activation:** Similar to above, with

$$\hat{x}_{k+1} = \sigma \left( W^{(0)} x_k + W^{(1)} x_{k-1} + \cdots + W^{(p-1)} x_{k-p+1} + b \right), \sigma = \exp$$

**Cross correlation:** Recall that $X_k = [x_{k-H+1} \quad \cdots \quad x_k] \in \mathbb{R}^{N \times H}$ denotes activity of $N$ neurons across $H$ time steps. Let $x^{(i)}, x^{(j)} \in \mathbb{R}^H$ denote the $i$-th and $j$-th neurons' activity across $H$ time steps. For simpliticity, let $a = x^{(i)} [\tau :], b = x^{(j)} [: -\tau]$. Cross correlation with time delay $\tau$ is computed as

$$r^{i \leftarrow j} = \frac{a^\top b}{\|a - \overline{a}\| \|b - \overline{b}\|}$$

We used $\tau = 1$ for inferring connectivity in both simulation and read data.

**Covariance:** Similar to above, let $x^{(i)}, x^{(j)} \in \mathbb{R}^H$ denote the $i$-th and $j$-th neurons' activity across $H$ time steps. Covariance between neuron $i$ and $j$ is defined as

$$c^{i \leftarrow j} = c^{j \leftarrow i} = \frac{1}{H-1} \Sigma_{k=1}^H x_k^{(i)} x_k^{(j)}$$

**Mutual information:** Computed using the Python package PyInform.mutualinfo.

**Transfer entropy:** Computed using the Python package PyInform.transferentropy.

### A.6.2 NONSTATIONARY CONNECTIVITY BASELINES FOR NEURAL DATA (SEC 4.4)

**Low-tensor-rank RNN (LtrRNN):** LtrRNN is designed to model trial-varying neural dynamics by capturing low-dimensional changes in connectivity over time. We split continuous neural activity data into overlapping sliding windows, treating each window as a distinct trial, which allows LtrRNN to learn temporal variations in connectivity patterns. The dimension of the network is defined by the number of neurons. To fit LtrRNN, we followed the code provided by its authors https://github.com/arthur-pe/LtrRNN. After fitting, we extracted the trial-specific weight matrices from LtrRNN and compared them with the attention maps learned by NetFormer.

**Autoregressive Hidden Markov Model (AR-HMM):** AR-HMM extends the traditional HMM by incorporating state-specific autoregressive dynamics, where each state has its own unique autoregressive model to describe the temporal dynamics of neural activity. AR-HMM is able to infer the latent states of the system and estimates the connectivity matrices associated with each state. However, a limitation of AR-HMM is the need to predefine the total number of states. To fit AR-HMM, we followed the code at https://github.com/lindermanlab/ssm.

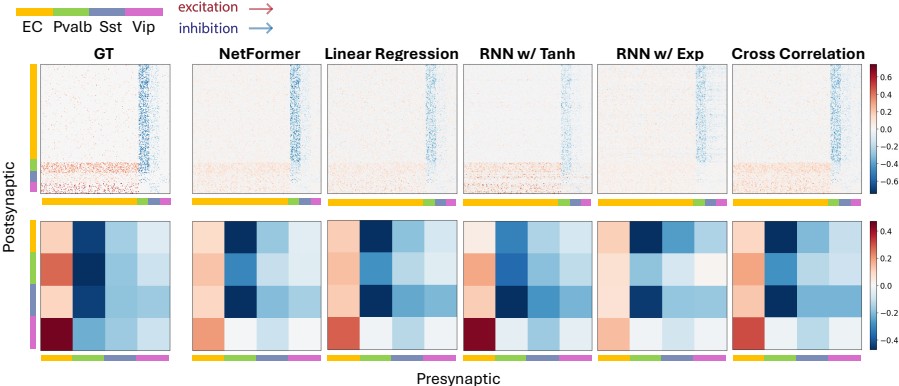

**Figure 11: Connectivity-constrained simulation with sigmoid activation.** Visualization of ground truth and inferred connectivity matrices at both individual-neuron level and cell-type level. A positive linear transformation has been applied to standardize all matrices to the same range for better visualization.

## A.7 MORE ON CONNECTIVITY-CONSTRAINED SIMULATION (SEC 4.2, 4.3)

### A.7.1 CONNECTIVITY-CONSTRAINED SIMULATION WITH SIGMOID NONLINEARITY

To test the robustness of NetFormer towards different nonlinear activations in the connectivity-constrained simulation (Section 4.2), we replaced the tanh activation with sigmoid activation and used NetFormer, along with other baselines, to reconstruct connectivity at both neuron-level and cell-type level. Table 5 presents quantitative comparisons of the inferred connectivitives with ground truth, averaged over five random seeds. Qualitative comparison is in Figure 11. NetFormer shows competitive performance, and outperforms other baselines at neuron-level ($N \times N$) connectivity inference.

|  |  |  | NetFormer | linear regression | RNN w/ tanh | RNN w/ exp | cross correlation | covariance | mutual information | transfer entropy |
|---|---|---|---|---|---|---|---|---|---|---|
| simulation | connectivity $N \times N$ | Pearson | **0.834**±0.006 | 0.765±0.002 | 0.772±0.001 | 0.679±0.002 | 0.829 | -0.022 | 0.447 | 0.329 |
|  |  | Spearman | **0.508**±0.005 | 0.482±0.002 | 0.485±0.001 | 0.454±0.000 | 0.512 | -0.006 | 0.201 | 0.223 |
|  | connectivity $K \times K$ | Pearson | 0.880±0.003 | 0.904±0.002 | **0.940**±0.001 | 0.803±0.002 | 0.921 | -0.438 | 0.350 | 0.304 |
|  |  | Spearman | 0.860±0.002 | 0.862±0.003 | **0.886**±0.000 | 0.788±0.011 | 0.856 | -0.299 | 0.104 | 0.340 |

**Table 5: Connectivity-constrained simulation with sigmoid activation.** Simulation data has ground truth for both neuron-level ($N \times N$) and cell type-level ($K \times K$) connectivities. Connectivity inference is assessed using Spearman's and Pearson's correlation coefficients. Results of connectivity inference using mutual information, transfer entropy, and Granger causality are assessed by comparing against the absolute values of ground truth connectivity.

### A.7.2 GRANGER CAUSALITY TEST

We performed Granger causality tests on connectivity-constrained simulation data with both tanh and sigmoid activations. To create a binary ground-truth neuron-level connectivity matrix, we assigned a value of 1 to all nonzero entries and 0 to zero entries. We then conducted Granger causality tests on pairwise neural activities to generate a matrix of test statistics, which is then min-max normalized to between 0 and 1. To compare with Granger causality on inferring binary connectivity, we also performed min-max normalization on the time-averaged neuron-level attention matrix. We evaluated the performance using AUROC. NetFormer outperforms Granger causality in both simulations (Figure 12).

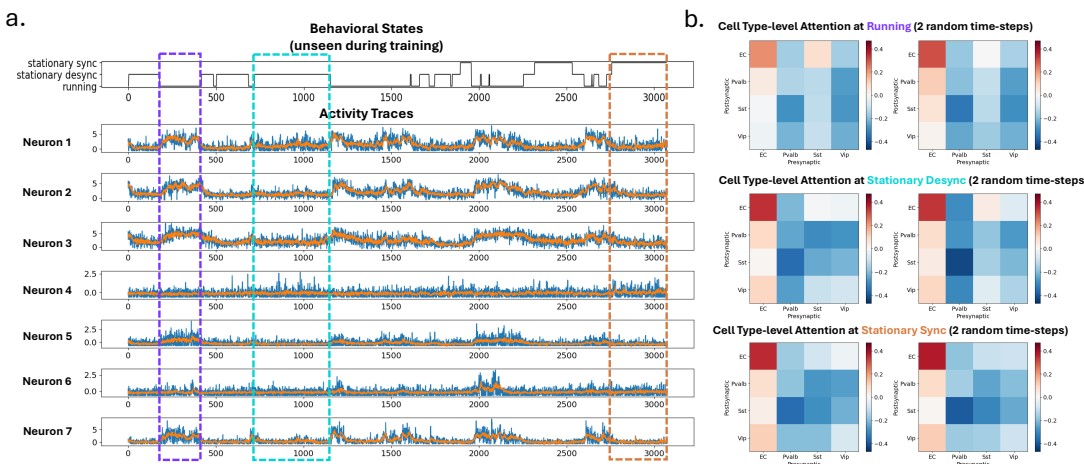

**Figure 12: NetFormer for binary connectivity recovery.** Visualization of NetFormer and Granger causality results for inferring binary connectivity (presence or absence of connections) in connectivity-constrained simulation data.

**Figure 13: a.** Visualization of neural data and behavioral states for an example session. Behavioral states are plotted in the top row. Predicted activity traces from NetFormer are shown in orange, while measured activity traces are shown in blue. NetFormer predictions effectively capture overall patterns of neural activity. **b.** Three blocks of time from **a.** are selected, each representing a different state. NetFormer attention weights at two timesteps randomly selected within each block are visualized.

## A.8 NETFORMER FOR NONSTATIONARY CONNECTIVITY INFERENCE ON NEURAL DATA (SEC 4.4)

We visualized neural activity traces, mouse behavioral states, and cell-type level attention weights for each state in Figure 13. While behavioral states are not provided as model inputs, they can be inferred using unsupervised methods such as PCA on NetFormer's attention weights. We also benchmakred NetFormer with LtrRNN and AR-HMM on state-dependent connectivity inference, as shown in Table 6. The inferred connectivities are compared against the patch-clamp experimental result.

## A.9 NETFORMER APPLIED TO NETWORKS WITH SPURIOUS CORRELATIONS (SEC 4.3)

Das & Fiete (2020) shows that connectivity inference can be obscured by spurious correlations arising from strong recurrent connections. To assess NetFormer's resilience to such spurious correlations, we performed two experiments on simulated neuron populations with strong recurrent connections. In the first experiment, we generated continuous activity from the simulated network, where NetFormer is directly applicable. In the second experiment, we generated spiking activity from the simulated network, which follows Das & Fiete (2020) more closely but requires modifying the objective function of NetFormer. In both experiments, we simulated neuron populations of size 100. The population connectivity matrices in both experiments are constructed in the same way as Das & Fiete

| | | | NetFormer | LtrRNN | AR-HMM |
|---|---|---|---|---|---|
| **in-vivo recording** | Running K × K | Pearson | **0.591**±0.204 | 0.007±0.182 | 0.199±0.027 |
| | | Spearman | **0.601**±0.209 | -0.036±0.189 | 0.274±0.048 |
| | Stationary Desync K × K | Pearson | **0.662**±0.176 | -0.003±0.181 | -0.320±0.012 |
| | | Spearman | **0.723**±0.173 | -0.057±0.174 | -0.206±0.017 |
| | Stationary Sync K × K | Pearson | **0.713**±0.148 | 0.000±0.186 | 0.145±0.041 |
| | | Spearman | **0.767**±0.151 | 0.069±0.175 | 0.151±0.009 |

Table 6: **State-dependent connectivity inference.** NetFormer attention maps and LtrRNN weight matrices are grouped by state labels and averaged for each state. The inferred connectivity for each state is compared against Postsynaptic Potential (PSP) resting state amplitude obtained from patch-clamp experiments.

(2020) with a local Mexican hat profile. Specifically, the connection weight between neurons $i$ and $j$ is computed as

$$w_{ij} = e^{-d_{ij}^2/2\sigma_1^2} - ae^{-d_{ij}^2/2\sigma_2^2}, \tag{25}$$

where $d_{ij}$ is the distance between neurons $i$ and $j$ (in units of neurons), $\sigma_1 = 6.98$, $\sigma_2 = 7$, and $a = 1.0005$.

### A.9.1  NETWORK WITH CONTINUOUS ACTIVITY

The network model used to generate the data is similar to equation 9, with modifications to the feed-forward inputs $b$ and incorporation of the recurrent weight strength $r$. Specifically, Gaussian noise is injected into the feed-forward inputs following Das & Fiete (2020):

$$\boldsymbol{x}_{k+1} = \tanh(r\boldsymbol{W}\boldsymbol{x}_k + \boldsymbol{b}(1 + \boldsymbol{\epsilon_k})). \tag{26}$$

We conducted similar experiments as in Das & Fiete (2020) by varying the recurrent weight strength $r$. The corresponding connectivity inference results are shown in Figure 14. When the recurrent weight strength is large (e.g. $r = 35$), spurious correlations between neurons appear, as seen in the synchronized activity and noise correlation. On the other hand, when the recurrent weight strength is small (e.g. $r = 5$), activity is mainly driven by noise rather than recurrent connections, making it hard to infer connectivity by fitting activity. In both extremes, NetFormer will struggle to capture the underlying ground-truth connectivity, as reflected by lower correlations with the ground truth. That being said, NetFormer still recovers some patterns of the ground truth, better than what noise correlation alone can capture.

### A.9.2  NETWORK WITH SPIKING ACTIVITY

To adapt NetFormer for modeling spiking activity, we take the exponential of NetFormer's output as the mean rate of the Poisson distribution for spike generation. The spike generation process is constructed in the same way as Das & Fiete (2020), which we describe briefly below, and we refer the readers to the original paper for more details.

Let $\boldsymbol{s}_k, \boldsymbol{g}_k, \boldsymbol{\sigma}_k$ denote the synaptic activation, neural input, and emitted spikes of the neuron population at timestep $k$, respectively. $\boldsymbol{g}_k$ includes both inputs from other neurons and externally injected inputs $\boldsymbol{b}_k$; that is,

$$\boldsymbol{g}_k = r\boldsymbol{W}\boldsymbol{s}_k + \boldsymbol{b}_k. \tag{27}$$

At timestep $k$, the $i$-th neuron will emit a spike, that is, $\boldsymbol{\sigma}_k^{(i)} = 1$, if the input it receives exceeds a certain threshold, that is, $\boldsymbol{g}_k^{(i)} > \Theta$.

To fit the NetFormer, at every timestep $k$, we have

$$\hat{\boldsymbol{g}}_k = \text{NetFormer}(\boldsymbol{s}_k, \dots, \boldsymbol{s}_{k-H+1}), \tag{28}$$

$$\hat{\boldsymbol{\lambda}}_k = \exp(\hat{\boldsymbol{g}}_k), \tag{29}$$

$$\hat{\boldsymbol{\sigma}}_k \sim Poisson(\hat{\boldsymbol{\lambda}}_k). \tag{30}$$

This setup is chosen to match the setup for GLM in Das & Fiete (2020) (see Equation 9 there). One minor difference is that, as model inputs, they used past spike trains convolved with a decaying

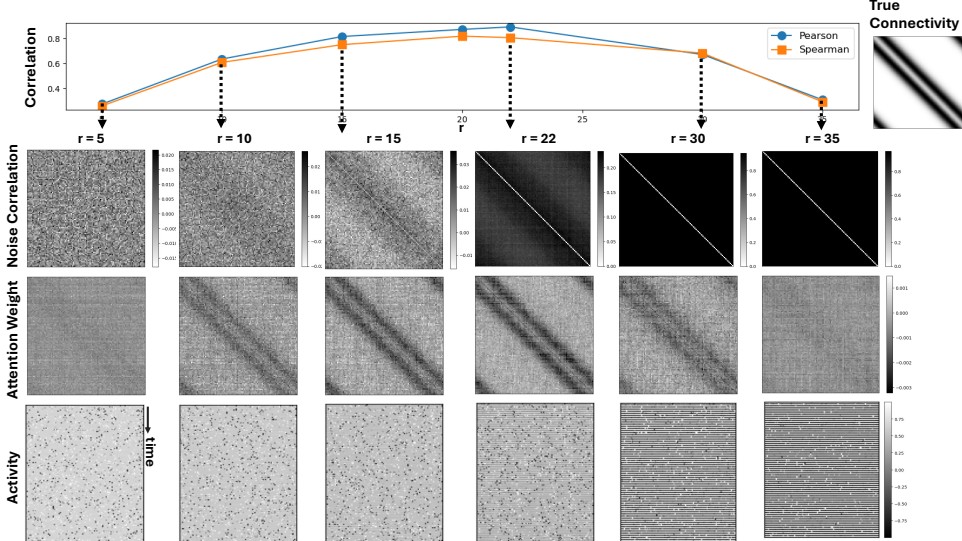

**Figure 14: Spurious correlation: network with continuous activity.** Correlation between attention weight as connectivity inference and ground-truth recurrent weights (top row), noise correlations (second row), attention weights (third row), and time-by-neuron activity traces (last row), at different recurrent weight strengths $r$.

exponential kernel, while we used the synaptic activation. Nonetheless, since their filter is truncated at a value much larger than the exponential decay time constant, it can be easily verified that these are actually very similar.

We followed the procedure described in Das & Fiete (2020) to generate population activity at different recurrent strength $r$. We followed the data generation code provided by the authors at https://github.com/FieteLab/neural-circuit-inference/blob/main/riken_demo.m. We adopted $r$ and spiking thresholds listed in https://github.com/FieteLab/neural-circuit-inference/blob/main/thresholds_pinned.mat. After the simulation starts, we collected activity after the initial 10,000 simulation timesteps in order for the network activity to stabilize. A total of 100,000 timesteps were collected, where the first 80,000 steps were used as the training set, and the rest 20,000 steps were used as the test set. Since this is a prototype study, the data volume we used here is much smaller than what is used in the orignal paper: their experiments are typically done on $10^8$ spikes, while there are on average $5.05 \times 10^4$ spikes in our training set across all $r$. Nevertheless, synchronized activity patterns and high noise correlation at large $r$ can also be observed in our training set (Figure 15).

For all $r$, we used NetFormer with $H = 1, N = 100, D = 100$. For $r = 0.0225$, $M = 40$, and $M = 100$ for the rest of $r$. All NetFormer models were trained to minimize the negative Poisson log likelihood loss on the training set, using batch size 32 for 100 epochs. We used learning rate 0.001 for $r = 0.0025, 0.005$, 0.002 for $r = 0.0075, 0.01, 0.0125, 0.015, 0.0175, 0.02, 0.025$, and 0.0005 for $r = 0.0225$. We used the average NetFormer attention across test set timesteps as the NetFormer-inferred connectivity. Each row in NetFormer-inferred connectivity is then rescaled to minimize the $l_2$ distance from the corresponding row in the ground-truth connectivity matrix, as also done in Das & Fiete (2020). Visually, Figure 15 shows that NetFormer-inferred connectivity can be affected by the presence of strong spurious correlation in neural activity, while being more resilient than noise correlation. Quantitively, we computed the Spearman and Pearson correlation coefficients between the off-diagonal entries of NetFormer-inferred connectivity and true connectivity. We also computed the inference error (the $l_2$ distance between the off-diagonal entries of true and inferred connectivity weights) as in Das & Fiete (2020). Consistent with their findings in Figures 2 and 3, we observe a U-shape curve in inference error as a function of $r$, and an inverted U-shape curve for correlation.

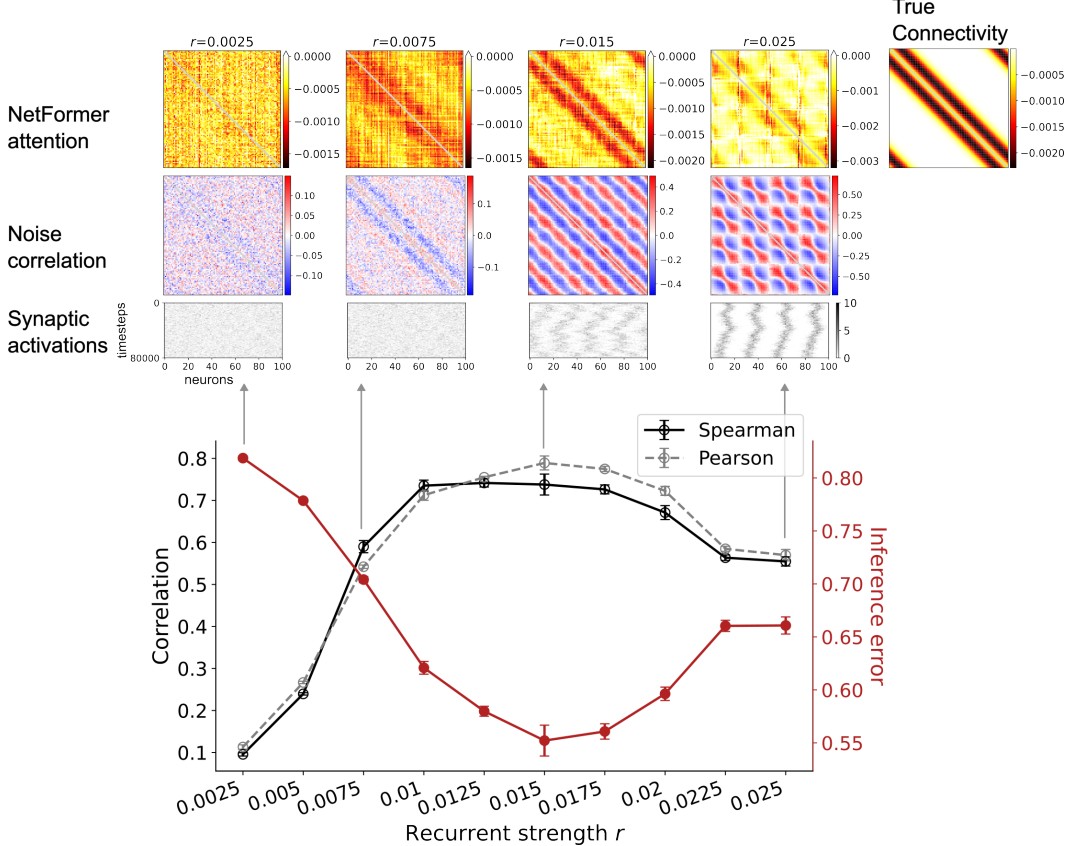

**Figure 15: Spurious correlation: network with spiking activity** Correlation and inference error between true and NetFormer-inferred connectivities. Errorbars: mean±std across three random initializations for NetFormer. In addition, rescaled NetFormer-inferred connectivity, noise correlation, and population synaptic activations at four example $r$ are visualized. For correlation and connectiviy matrices, colorbars indicate the scale of the off-diagonal entries, and the diagonal entries are masked in grey.

## A.10 IMPLEMENTAION DETAILS FOR FITTING CONNECTIVITY-CONSTRAINED SIMULATION AND NEURAL DATA (SEC 4)

### A.10.1 MODEL FRAMEWORK

Following Section 2, we train the NetFormer to predict $x_{k+1}$ based on $X_k = [x_{k-H+1} \quad \cdots \quad x_k] \in \mathbb{R}^{N \times H}$. To encode neuron identities, a learnable positional embedding matrix $E \in \mathbb{R}^{N \times M}$ is concatenated to $X$, giving $\tilde{X}_k = [X_k \quad E] \in \mathbb{R}^{N \times (H+M)}$. The queries $Q_k$ and keys $K_k$ are obtained through linear transformations of $\tilde{X}_k$, $Q_k = \tilde{X}_k W_Q \in \mathbb{R}^{N \times D}$, and $K_k = \tilde{X}_k W_K \in \mathbb{R}^{N \times D}$. NetFormer model is trained to predict the next time-step activity $x_{k+1}$, defined as

$$\hat{x}_{k+1} = A_k x_k + x_k = \phi(\frac{Q_k K_k^T}{\sqrt{D}})x_k + x_k = \frac{1}{\sqrt{D}}(\tilde{X}_k W_Q)(W_K^\top \tilde{X}_k^\top)x_k + x_k,$$

where $A_k$ is the self-attention that we want to use for inferring connectivity, $\phi$ is the attention activation. In the standard Transformer model Vaswani et al. (2017), softmax is used as the attention activation function, but here we set $\phi$ equal to identity for better interpretability. In fact, we experimented with different activation functions and empirically found that the identity activation yields the best results on recordings from the mouse cortex.

Although not used in the current experiments, it is possible to apply an additional linear transformation ($w_{out}$) on $X_k$ to accommodate neuronal dynamics that can depend on multiple previous timesteps,

that is,

$$\hat{x}_{k+1} = A_k(X_k w_{out}) + X_k w_{out}, w_{out} \in \mathbb{R}^{H \times 1}.$$

### A.10.2 NetFormer training and evaluation

We first assign each unique neuron across all sessions an ID, which is later used to track positional embedding for each unique neuron, because same neuron can be recorded in more than one session. Then, within each session, we construct samples with window size 200 in simulation and 60 in real data, and we make sure samples in one batch should come from the same session so that the dimensions can match.

We use the first $80\%$ timesteps in all sessions for training and the last $20\%$ timesteps for validation. The model is trained using MSE as the loss function, comparing the predicted activity for the next time step with the ground-truth activity. We employ early stopping criteria, ceasing training if there are 20 epochs without improvement, with a hard limit of 100 epochs maximum.

After training is complete, we calculate the attention for each sample in the dataset. For each session, we aggregate the attentions from all samples to compute a single time-averaged attention. Averaged attentions from all sessions are then transformed into a final cell-type level attention. We achieve this by aggregating attention values according to their corresponding presynaptic and postsynaptic cell types and dividing by the total count of such pairs.

We also extract positional embeddings from the trained model and utilize each neuron's unique ID to determine the neuronal embedding for every unique neuron. These embeddings are then used as features for logistic regression to classify neurons as either excitatory or inhibitory (Figure 5b).

For evaluation, we assess the inferred cell-type level connectivity against the Postsynaptic Potential (PSP) resting state amplitude obtained from patch-clamp experiments, which serves as the experimental ground-truth. Additionally, we evaluate the accuracy of the binary cell-type classification using experimental data from single-cell spatial transcriptomics, which provides a classification of neurons into excitatory and inhibitory types across all sessions.

### A.10.3 Evaluation metircs

We use python libraries and built-in functions for computing evaluation metrics.

For connectivity inference, we flatten the inferred 2-dimensional $N \times N$ or $K \times K$ connectivity matrix and grond-truth matrix.
**Pearson correlation:** scipy.stats.pearsonr()
**Spearman rank correlation:** scipy.stats.spearmanr().

For activity prediction, given the input matrix $\in \mathbb{R}^{B \times N \times H}$ for NetFormer and input matrix $\in \mathbb{R}^{B \times N}$ for RNN, where $B$ is the batch size, NetFormer outputs $\in \mathbb{R}^{B \times N \times 1}$ and RNN outputs $\in \mathbb{R}^{B \times N}$. We flatten the predicted activity and the grond-truth.
**MSE:** torch.nn.functional.mse_loss()
**Pearson correlation:** scipy.stats.pearsonr()
**$R^2$ :** sklearn.metrics.r2_score()

For binary classification, classifier predicts the probability for all neurons.
**Top-1 accuracy:** sklearn.metrics.accuracy_score()
**Area Under the Receiver Operating Characteristic (AUROC):** sklearn.metrics.roc_auc_score()

### A.10.4 Hyperparameters

In connectivity-constrained simulation data, for training NerFormer, we use history window size 100, embedding size 200, hidden dimension of query and key matrices is 300, learning rate $10^{-3}$, and batch size 32. For training RNN, we use $p = 1$, batch size 32, and learning rate $10^{-3}$.

In neural recording, for training NetFormer, we use history window size 60, embedding size 30, hidden dimension of query and key matrices 90, learning rate $10^{-3}$, and batch size 32. For training RNN, we use $p = 1$, batch size 32, and learning rate $10^{-4}$.

We use PyTorch Paszke et al. (2017) and PyTorch Lightning Falcon & The PyTorch Lightning team (2019) for model development and training, and Adam as the optimizer.

### A.10.5 PSEUDOCODE

We train NetFormer and extract attentions and positional embeddings for connectivity inference and binary cell-type classification. The pseudo code for model training , connectivity inference and cell-type classification is provided as follows:

```
NetFormer(x, neuron_ids):
    if constraint == True:
        cell_type_level_mean = parameters(num_cell_type, num_cell_type)
        cell_type_level_var = parameters(num_cell_type, num_cell_type)

    embeddings = embedding_table(neuron_ids)
    input = layer_norm(concat(x, embeddings))
    x, embeddings = input[:, :, :T], input[:, :, T:]

    dim_x, dim_e = x.shape[-1], embeddings.shape[-1]
    scale = (dim_x + dim_e) ** -0.5

    logits = input @ W_Q_W_KT @ input.T
    logits = logits * scale

    if activation == softmax:
            attention = softmax(logits)
        elif activation == sigmoid:
            attention = sigmoid(logits)
        elif activation == tanh:
            attention = tanh(logits)
        elif activation == none:
            attention = logits

    output = layer_norm(attention @ x + x)

    if out_layer == True:
        # linear_out is a lienar transformation from dimension T to 1
        output = linear_out(output)
        return output, attention
    else:
        # Use the last column as prediction
        return output[:, :, -1], attention

NetFormer_Training(all_samples):
    all_inputs, all_neuron_ids, all_GT_targets = all_samples
    model = NetFormer()
    optimizer = Adam(model, learning_rate)

    all_predictions, all_attentions = model(all_inputs, all_neuron_ids)

    prediction_loss = MSE(all_predictions, all_GT_targets)
    loss = prediction_loss

    optimizer.zero_grad()
    loss.backward()
    optimizer.step()
```

```
Connectivity_Inference(all_samples, trained_NetFormer, GT_connectivity):
    all_inputs, all_neuron_ids, all_GT_targets = all_samples
    all_predictions, all_attentions = trained_NetFormer(all_inputs, all_neuron_ids)

    avg_attention = mean(all_attentions, axis=0)

    pearson_corr = pearsonr(GT_connectivity, avg_attention)
    spearman_corr = spearmanr(GT_connectivity, avg_attention)

Cell_Type_Classification(trained_NetFormer, neuron_ids, cell_types):
    embeddings = trained_NetFormer.embedding_table(neuron_ids)

    X_train = embeddings[TRAIN_idx]
    y_train = cell_types[TRAIN_idx]
    X_test = embeddings[TEST_idx]
    y_test = cell_types[TEST_idx]

    # Train classifier
    classifier = LogisticRegression.fit(X_train, y_train)
    # Test on test set
    y_pred = classifier.predict(X_test)
```

### A.11 COMPUTE RESOURCES

Model training on simulated systems in Section 3 was done on a MacBook Pro with Apple M1 chip. Using the NVIDIA A100 GPU, NetFormer model trained on connectivity-constrained simulation data took about 10min. On neural recording, NetFormer model trained on the subject SB025 took about 20min, which requires at least 30 GB of RAM and 16 GB of GPU memory.

