# OpenReview forum: "NetFormer: An interpretable model for recovering dynamical connectivity in neuronal population dynamics"
_ICLR.cc/2025/Conference — ICLR 2025 Spotlight_

### Official Review · Reviewer_ovPw · 2024-10-26

**Soundness:** 3
**Presentation:** 2
**Contribution:** 3
**Rating:** 8
**Confidence:** 4

**Summary:**

This paper introduces a model (NetFormer) for estimating the time-varying connectivity between neurons given access to recordings of their activity. At each timepoint, the approach maps the recent history of neural activity to an estimated connectivity matrix based on a linearized attention mechanism. The authors apply their method on a combination of synthetic tasks and real-world datasets, demonstrating that it can successfully capture nonstationary effects, as well as recapitulate experimental measures of connectivity in realistic settings. They show that their approach outperforms several baselines, both in predicting activity, as well as in estimating connectivity matrices.

**Strengths:**

Drawing inspiration from the attention mechanism used in transformers, the authors propose a method for estimating connectivity weights when the underlying dynamics are nonstationary. The proposed method is simple and interpretable, and is shown to outperform baselines with static connectivity, especially in synthetic tasks where there is good control over ground truth. Further, they demonstrate the utility of their approach for auxiliary tasks, such as cell type inference from learned positional embeddings. The majority of methods for connectivity inference in the neuroscience literature are not well-suited for the nonstationary setting, so this work does indeed address an important question.

**Weaknesses:**

The estimated connectivity at time $k$ is constrained to be of the form $\mathbf{A}_k = \tilde{\mathbf{X}}_k\mathbf{W}_Q\mathbf{W}_K^\top \tilde{\mathbf{X}}_k^\top$. This can be intepreted as a kind of generalized covariance of the recent history of neural activity, where $\mathbf{W}_Q\mathbf{W}_K^\top$ performs some form of temporal mixing/weighting. Given this, I suspect that the proposed method may be particularly susceptible to spuriously estimating connectivity between unconnected neurons that nonetheless have highly correlated activity (as discussed in Das & Fiete, 2020), even in a time-lagged sense. It is unclear to me whether any of the experiments presented test this scenario. One possibility is to test how NetFormer performs on a synthetic network with Mexican hat connectivity and white driving noise, as done in Das & Fiete, 2020.

My biggest reservation is that the baselines used in the experiments were rather weak. In particular, no other baselines that could account for time-varying connectivity were compared to. While admittedly the vanilla static RNNs used as baselines are standard in computational neuroscience, one could easily imagine a time-varying linear model of the form $x_{k+1} = W_kx_{k}+b$, where $W_k$ is estimated via OLS over a time window $[k-H,...k]$; such models are common in the signal processing literature. One could also imagine a nonlinear version $x_{k+1} = \tanh(W_kx_{k}+b)$.

**Questions:**

1. The claimed performance in estimating experimentally measured cell type-averaged postsynaptic potential (Table 1) is quite remarkable, yet I was left confused on some of the details here. For example, the fitted model appears to predict different cell type-averaged activity for different behavioral "states" (Fig. 5),  yet it appears there is a single cell type-averaged ground truth connectivity derived from patch clamp experiments that all models in Table 1 are being compared to. Is it simply the case that "connectivity" measured under patch-clamp conditions happens to match an average over these "states"?  It would be useful to help further evaluate the claimed performance on this dataset if the authors provided a supplementary figure showing additional details on these fits, as done in other figures (e.g. fitted connectivity vs the ground truth "connectivity" for NetFormer and the various baselines, fitted activity traces, etc.)

2. Minor: the MSE between the average connectivity and the ground truth is used throughout as a performance metric for NetFormer. This is sensible when average connectivity is used as the final readout of estimated connectivity. Does this metric significantly differ from the average MSE between the time-varying connectivity and the ground truth? This is arguably a more accurate measure of how much the model deviates from the ground truth, as deviations at all times should be penalized.

---

> ### Author Response · Authors · 2024-11-22
> **Thanks for your constructive feedbacks**
>
> Thank you for your thoughtful review! We have revised our manuscript (with revisions marked in purple). We hope the revision and the following responses could address some of your concerns:
>
> - **MSE as performance metric:** Thanks for the question. We wanted to clarify that we used MSE to evaluate our dynamics prediction, not connectivity inference. During training, we have used the MSE between the true and predicted next-step activity to guide model parameters optimization. In Table 1, the MSE metric is for evaluating next-step activity prediction on the test set, instead of evaluating connectivity inference. Please refer to Table 1 caption for more details.
>
> - **Spurious correlation:** Thanks for the insightful question. We compared NetFormer with cross-correlation (Table 1, Figure 5) for connectivity strength inference and with Granger causality (Appendix A7.2, Figure 12) for binary connectivity inference. In both cases, NetFormer demonstrated better performance, leading us to conclude that our model can capture effective connection beyond mere spurious correlation. Meanwhile, we would also like to highlight that we compared the connectivity estimated from our method with the cell-type level connectivity measured from the patch-clamp experiment. It involves stimulating one cell and recording from another, thereby establishing causal connectivity in neural circuits. Additionally, the partial observation experiment in Figure 5 also shows the robustness of our approach to hidden confounding variables. We appreciate the reviewer's suggestion to evaluate NetFormer on alternative datasets. We believe this is a valuable direction for future work.
>
> - **Baselines accounting for time-varying connectivity:** Thanks for the suggestions. We have added two baselines that account for trial-(state-) varying connectivity: LtrRNN ([1] Pellegrino et al., 2023, as suggested by reviewer 9MUd) and AR-HMM ([2], [3]) in section 4, Figure 6, and Appendix A8. Specifically,  LtrRNN models trial-varying connectivity using a low rank tensor $\mathbf{W} \in \mathbb{R}^{N \times N \times K}$, where $K$ denotes the number of trials. AR-HMM assumes that there are discrete latent states switching underlying the observed activity, and each state admits unique dynamics through a different connectivity matrix. We observed that compared to weights inferred by LtrRNN, PCA on NetFormer-inferred weights yields a cleaner separation between the running and stationary states. The states inferred by AR-HMM for each timestep shows some similarity with true states but has higher noise. Moreover, compared to LtrRNN and AR-HMM, inferred connectivities from NetFormer’s attention are in better agreement with the cell-type level patch-clamp measurements. We would also like to point out that NetFormer is more efficient than LtrRNN: NetFormer uses shared parameters across time/trials, while LtrRNN trains trial-specific parameters, leading to an increase in parameters as the number of trials grows. Also, discrete state change modeled by AR-HMM is not ideal for capturing continuous connectivity changes, and heavily relies on the user-specified number of states.
>
> - **A single cell type-averaged ground truth connectivity:** We included additional visualizations of neural activity traces, mouse behavioral states, and cell-type level attention weights for each state within a block of time in Figure 13 of Appendix A8. We observed that the inferred intra-state connectivity is more similar compared to connectivity across different states.
> We would like to clarify that the patch-clamp cell-type connectivity data and the multi-modal neural activity data were obtained from separate experiments, meaning that there does not exist an exact ground-truth connectivity structure underlying each behavioral state or timestep. Nevertheless, even for state-dependent connectivity inference (Table 6 in Appendix A8), we believe that the patch-clamp experimental results still provide a reasonable reference, with the following reasoning: As the neural data was collected during spontaneous activity, even though there are some periods with high locomotion (as in the running state), there is no learning or perturbations that could alter the wiring of the neural population drastically. Meanwhile, our results are also consistent with the findings of cell-type specific state-dependent connectivity reported in Fu et al. 2014: VIP cells are more activated during locomotion, while SST cells are more inhibited, suggesting “a circuit in which VIP cells increase activity of neighboring excitatory cells by inhibiting their inhibitory input from SST cells” (Fu et al. 2014). In Fig 6d top row, NetFormer-inferred attention shows that the presynaptic VIP column (rightmost column) is darker (more inhibition) in the running state compared to the stationary states and the presynaptic SST column (second column from the right) is lighter (less inhibition).

---

> > ### Comment · Reviewer_ovPw · 2024-11-25
> >
> > - I appreciate the authors' revisions, especially the inclusion of baselines that handle nonstationarity, which clarify the contribution of the proposed method. I still maintain that fitting to spurious correlations is a potential problem with the proposed method, and am not convinced that the causality tests performed rule that out. For example, a network with random gaussian weights will not produce the strong activity correlations that lead to fitting spurious connections. While the tested synthetic networks were indeed more structured than that, I would still like to see how the proposed method performs on networks specifically known to create such problems (as also requested by reviewer 9MUd).
> >
> > - I also remain impressed by the performance in recovering patch-clamp measured activity listed in Table 6. I appreciate that the authors have now included the code they used for experiments, and wanted to do a quick sanity check of those findings. However, I could not find the implementation of the baseline methods (e.g. `baselines.py` doesn't include the baseline implementations or evaluations). Apologies for the short notice, but could the authors clarify where that is and/or update the code to include that?
> >
> > Overall though, given that the authors have agreed to include experiments on networks that produce strong activity correlations, I am inclined to raise my score once I receive clarification on the second point above.

---

> > > ### Author Response · Authors · 2024-11-27
> > >
> > > **Spurious correlation:** Thanks for the suggestion and your positive feedback on our work. In Appendix 9, we conducted two experiments on simulated neuron populations with strong recurrent connections: one generating continuous activity; and the other generating spiking activity, which follows Das & Fiete (2020) [1] more closely but requires modifying NetFormer’s objective function. We experimented with different values of recurrent strength in both networks. Adjusting the recurrent strength alters the network’s regime, leading to stronger or weaker levels of spurious correlations. In both simulations, we showed that NetFormer shows some resilience to spurious correlations, while can also struggle when the recurrent weights are too strong or too weak, similar to other methods studied in Das & Fiete (2020), including the GLM [2], logistic regression [3], and the Ising model [4].
> > >
> > > **References**
> > >
> > > [1] Das, A., & Fiete, I. R. (2020). Systematic errors in connectivity inferred from activity in strongly recurrent networks. Nature Neuroscience, 23(10), 1286-1296.
> > >
> > > [2] Pillow, J. W., Shlens, J., Paninski, L., Sher, A., Litke, A. M., Chichilnisky, E. J., & Simoncelli, E. P. (2008). Spatio-temporal correlations and visual signalling in a complete neuronal population. Nature, 454(7207), 995-999.
> > >
> > > [3] Lee, S. I., Lee, H., Abbeel, P., & Ng, A. Y. (2006, July). Efficient l~ 1 regularized logistic regression. In Aaai (Vol. 6, pp. 401-408).
> > >
> > > [4] Roudi, Y., Tyrcha, J., & Hertz, J. (2009). Ising model for neural data: model quality and approximate methods for extracting functional connectivity. Physical Review E—Statistical, Nonlinear, and Soft Matter Physics, 79(5), 051915.
> > >
> > >
> > > **Code for spurious correlation experiment:** We also updated our code to include two spurious correlation experiments in supplementary materials.
> > >
> > > **Code for nonstationary baseline implementation:** We appreciate your interest in our work. To clarify, we did not implement LtrRNN and AR-HMM ourselves. Instead, we directly used the publicly available code from the following repositories: LtrRNN https://github.com/arthur-pe/LtrRNN and AR-HMM https://github.com/lindermanlab/ssm. These two repositories are also referenced in Appendix 6.2, where we describe the nonstationary baselines. To assist a quick run on the neural data (Bugeon et al., 2022, Nature), we have provided two notebooks within the updated code. Before running the notebooks, please first clone the two GitHub repositories (LtrRNN and AR-HMM) and install their dependencies accordingly. We are actively working on organizing the code for all experiments and evaluations, and a more comprehensive version will be made publicly available.

---

> > > > ### Comment · Reviewer_ovPw · 2024-11-27
> > > >
> > > > I greatly appreciate the new set of experiments and figures on fitting to strongly correlated activity, as well as the clarifications regarding implementations. I believe that the inclusion of a more comprehensive set of baselines, evaluations and analyses following this discussion period has improved the level of contribution considerably. As such, I have raised my score accordingly.

---

> ### Author Response · Authors · 2024-11-22
>
> **References**
>
> [1] Pellegrino, A., Cayco Gajic, N. A., & Chadwick, A. (2023). Low tensor rank learning of neural dynamics. Advances in Neural Information Processing Systems, 36, 11674-11702.
>
> [2] Fox, E., Sudderth, E., Jordan, M., & Willsky, A. (2008). Nonparametric Bayesian learning of switching linear dynamical systems. Advances in neural information processing systems, 21.
>
> [3] Linderman, S., Johnson, M., Miller, A., Adams, R., Blei, D., & Paninski, L. (2017, April). Bayesian learning and inference in recurrent switching linear dynamical systems. In Artificial intelligence and statistics (pp. 914-922). PMLR.

---

### Official Review · Reviewer_ipsn · 2024-11-04

**Soundness:** 3
**Presentation:** 2
**Contribution:** 3
**Rating:** 6
**Confidence:** 4

**Summary:**

The paper introduces NetFormer, a Transformer-based model designed to capture dynamic and nonstationary connectivity structures in neural systems. Through linearized attention mechanisms that bypass the softmax constraint, NetFormer encodes neuronal interactions that evolve over time. The model's performance is tested on synthetic data and real neural recordings from the mouse visual cortex, showing promising results in predicting neural activity and recovering cell-type connectivity.

**Strengths:**

1.  The conceptualization of linear Transformers for dynamically evolving connectivity is well-conceived, bridging gaps left by traditional static models. It provides a valuable alternative by proposing a method to infer state-dependent connectivity in neural circuits, which addresses the challenge of nonstationarity.
2.  The paper demonstrates the model's robustness by evaluating it on both simulated data and real biological recordings, including calcium imaging. The comparison with an oracle baseline further highlights the method's effectiveness and provides a strong benchmark for its performance across different neural contexts.

**Weaknesses:**

1. Model Interpretability in Practice: Although the model is interpretable in its mathematical formulation, it remains challenging to track how specific interactions are captured in real neural data, which may affect its transparency in biological applications. Visualization of the attention weights and their correspondence to neural trajectories could enhance interpretability.

2. Limited Comparison with Advanced Baselines: The model's comparisons primarily involve classical methods or simpler RNN architectures. Recent models like Neural ODEs and dynamic GNNs, which have been shown to capture temporal and spatial interactions effectively, are not included. This absence weakens the claims of state-of-the-art performance

3. Positional Encoding: The rationale behind applying positional encoding across the spatial dimension of time series data from multiple neurons is unclear and warrants further clarification. While positional encoding is typically well-suited for distinguishing between different time steps in a temporal sequence, the authors should explain the motivation and justification for using it in the spatial context of calcium imaging data. If the encoding is based on the physical locations of neurons, it should be specified, though this does not appear to be the case in this paper. Additionally, the reasoning for using positional embeddings to categorize neurons as excitatory or inhibitory requires a more detailed explanation.

4. Dataset Specificity and Transferability: The real neural data focuses on calcium imaging with lower temporal resolution than electrophysiological recordings. This choice may limit the model's applicability in higher-temporal-resolution neural data, which would be better suited for capturing fast synaptic interactions. Additionally, the use of positional embeddings in calcium imaging data needs further justification, as it might not correlate well with physical neuron placement.

5. Reproducibility: The authors have not provided any code repository or made their scripts publicly available, which raises concerns about the reproducibility of their results. Although pseudocode is included in the supplementary materials, it does not fully address the need for accessible implementation details to enable independent verification of their findings.

**Questions:**

1. Could you incorporate visualizations that link the learned attention weights to specific neural trajectories or connectivity structures, perhaps using case studies or examples from the experimental dataset?

2. Have you considered including comparisons with more recent and relevant models, such as Neural Data Transformers, Neural ODEs, or temporal GNNs? Additionally, would exploring ablations with models specifically designed for connectivity tasks, like GLMs with state-switching mechanisms, help further validate NetFormer’s performance claims?

3. Would it be beneficial to use alternative metrics for connectivity evaluation, such as causality metrics or time-lagged correlations, to provide more biologically relevant insights?

4. Could you clarify the rationale behind applying positional encoding across the spatial dimension of time series recorded from multiple neurons? While positional encoding is typically justified for temporal sequences, how does it apply to the spatial dimension of calcium imaging data? Additionally, what is the reasoning for using these positional embeddings to classify neurons as excitatory or inhibitory?

Overall, if the authors are able to address some of these concerns I would be happy to increase my initial score accordingly.

---

> ### Author Response · Authors · 2024-11-22
> **Thanks for your constructive feedbacks**
>
> Thank you for your thoughtful review! We have revised our manuscript (with revisions marked in purple). We hope the revision and the following responses could address some of your concerns:
>
> - **Visualization of the attention weights and neural trajectories:** We have added visualizations of neural activity traces, mouse behavioral states, and cell-type level attention weights for each state within a block of time in Figure 13 of Appendix A8. We observed that the inferred intra-state connectivity is more similar compared to connectivity across different states.
>
>
> - **More recent and relevant models:** We appreciate the reviewer’s input in brainstorming relevant baselines. We have added two new baselines capable of capturing nonstationary connectivity: LtrRNN ([1] Pellegrino et al., 2023, as suggested by reviewer 9MUd) and AR-HMM ([2], [3]) in section 4, Figure 6, and Appendix A8. Specifically,  LtrRNN models trial-varying connectivity using a low rank tensor $\mathbf{W} \in \mathbb{R}^{N \times N \times K}$, where $K$ denotes the number of trials. AR-HMM assumes that there are discrete latent states switching underlying the observed activity, and each state admits unique dynamics through a different connectivity matrix. We found that NetFormer is able to capture connectivity changes between behavioral states more clearly. Moreover, compared to LtrRNN and AR-HMM, NetFormer-inferred connectivities from neural recordings are in better agreement with the cell-type level patch-clamp measurements (Table 6 in Appendix A.8). We also want to comment that NetFormer is more efficient than LtrRNN: for a given neuronal population, NetFormer uses shared parameters across time/trials, while LtrRNN trains trial-specific parameters, leading to an increase in parameters as the number of trials grows. For AR-HMM, its discrete state switching mechanism makes it less suitable for capturing continuous connectivity changes, and its state discovery heavily relies on the user-specified number of states.
>
> On the other hand, the following methods seem less applicable to our focus:
>   - **Neural Data Transformers (NDT):** Using multi-layer, multi-headed transformer models, NDT can produce accurate dynamics forecasts. However, for models with more layers and parameters, their attention weights can be hard to identify or interpret. Moreover, with the softmax activation applied in their attention computation, NDT’s attentions cannot model that the effect of one neuron on another can be either excitatory or inhibitory, whereas NetFormer employs a linearized attention mechanism without softmax to account for this fact. Resolving the ambiguity in interpretability of NDT’s attention is beyond the scope of the current study.
>   - **Temporal GNNs:** While some GNN models are able to describe both spatial and temporal relationships, they have two major limitations, as also pointed out in [4]: 1. They often require a pre-defined graph, while in the neuroscience setting, connectivity information in the neuronal population recording is rarely available. 2. The graph they learned is often static, which will not be able to account for the nonstationary connectivity that we are interested in modeling. Therefore, we believe that GNNs models are not suitable for our goal of discovering nonstationary connectivity from fitting dynamics.
>   - **Neural ODEs:** Neural ODEs can be a powerful model for fitting neural dynamics as shown in [5]. Nevertheless, their focus is not to recover dynamical neuronal connectivity, and thus we do not think it fits into the current study.
>
> - **Alternative metrics for connectivity evaluation, such as causality metrics or time-lagged correlations:** Thanks for the suggestions. In fact, one of our existing baselines, cross correlation, is exactly based on time-lagged correlations, and has been applied on both connectivity-constrained simulation and neural data, as shown in Table 1, Figure 5, Table 5, and Figure 11. We also added Granger causality (GC) on connectivity-constrained simulation as an additional baseline. As shown in Figure 12 and Appendix 7.2, NetFormer achieves better AUROC than GC in inferring binary connectivity. Meanwhile, we would like to emphasize that GC is commonly used for measuring *binary* connectivity (connected or not), but in both our simulations and neural data, the ground-truth connectivity has *signs (excitatory or inhibitory)* and *magnitude (strength)*, which cannot be captured by Granger causality.

---

> ### Author Response · Authors · 2024-11-22
>
> - **Positional encoding:** Thanks for the question and we are more than happy to clarify. The positional encodings are learnable parameters of NetFormer to reflect individual neuron’s identity, and are not based on the physical locations of recorded neurons. The original transformer paper [6] has argued the importance of positional embedding in transformer models: The transformer architecture treats input tokens in parallel, without any built-in mechanism to understand their positions in a sequence, and the positional embeddings enable the model to distinguish between tokens at different positions. In our case, recall that each token is the activity of one neuron. So we assign each neuron a unique learnable positional embedding vector, which can help represent the identity of each neuron within the population and help the model better understand their roles. Without positional embeddings, the model would treat each neuron identically. As for using position embeddings for cell-type decoding, it has been shown in [7] that learned positional embeddings can represent neuron identities and be used for cell-type classification.
>
> - **Dataset specificity and transferability:** We agree with the reviewer that compared to the current dataset, electrophysiological recordings can make a better dataset with higher temporal resolution. Nevertheless, given our focus on recovering cell-type level connectivity, currently we are not aware of any publicly available electrophysiological neural activity recordings with simultaneous cell-type labels and connectivity ground truth as an alternative to the current imaging dataset. We look forward to testing our model on such datasets once they are available.
>
> - **Code availability:** We have attached our code in the supplementary material. We will also make this code publicly available.
>
> **References**
>
> [1] Pellegrino, A., Cayco Gajic, N. A., & Chadwick, A. (2023). Low tensor rank learning of neural dynamics. Advances in Neural Information Processing Systems, 36, 11674-11702.
>
> [2] Fox, E., Sudderth, E., Jordan, M., & Willsky, A. (2008). Nonparametric Bayesian learning of switching linear dynamical systems. Advances in neural information processing systems, 21.
>
> [3] Linderman, S., Johnson, M., Miller, A., Adams, R., Blei, D., & Paninski, L. (2017, April). Bayesian learning and inference in recurrent switching linear dynamical systems. In Artificial intelligence and statistics (pp. 914-922). PMLR.
>
> [4] Grigsby, J., Wang, Z., Nguyen, N., & Qi, Y. (2021). Long-range transformers for dynamic spatiotemporal forecasting. arXiv preprint arXiv:2109.12218.
>
> [5] Kim, T. D., Luo, T. Z., Pillow, J. W., & Brody, C. D. (2021, July). Inferring latent dynamics underlying neural population activity via neural differential equations. In International Conference on Machine Learning (pp. 5551-5561). PMLR.
>
> [6] Vaswani, A. (2017). Attention is all you need. Advances in Neural Information Processing Systems.
>
> [7] Mi, L., Le, T., He, T., Shlizerman, E., & Sümbül, U. (2023). Learning time-invariant representations for individual neurons from population dynamics. Advances in Neural Information Processing Systems, 36, 46007-46026.

---

> > ### Comment · Reviewer_ipsn · 2024-11-26
> >
> > Thank you for your detailed responses and thoughtful revisions. I appreciate the effort you have put into addressing my concerns and incorporating suggestions from all reviewers. Based on the updates and clarifications provided, I believe the manuscript has improved in several key areas. The added visualizations, new baseline comparisons, and clarifications on methodological choices enhance the robustness and clarity of the manuscript. While some limitations remain inherent to the dataset and model scope, the authors have provided reasonable justifications and demonstrated a clear path forward.
> >
> > That said, I believe the manuscript can still benefit from improvements in writing and presentation. A more streamlined narrative and clearer explanations in certain sections (e.g. section 3) could further enhance the readability and accessibility of the paper, making its contributions more apparent to a broader audience.
> >
> > Based on these improvements and considering feedback from other reviewers, I am happy to increase my score.

---

> > > ### Author Response · Authors · 2024-11-27
> > >
> > > Thank you for the suggestion and for raising the score. We will make sure to work on improving writing and presentation in future revisions.

---

### Official Review · Reviewer_HFca · 2024-11-07

**Soundness:** 4
**Presentation:** 3
**Contribution:** 4
**Rating:** 8
**Confidence:** 4

**Summary:**

In order to overcome the shortcomings of static and non-interpretative models, the research presents NetFormer, a novel model for assessing dynamic connectivity in neural networks. In order to infer nonstationary and nonlinear connection, NetFormer employs a linearized attention mechanism. Without softmax restrictions, the model uses a time-dependent attention mechanism and treats brain activity across time as tokens to achieve interpretability, which makes it more biologically plausible. Spike-timing-dependent plasticity (STDP) simulations, synthetic data, and actual brain recordings are used to evaluate the model, which shows that it can accurately predict neural activity and capture dynamic neuronal connections.

**Strengths:**

Originality: A genuine gap in the present neuroscience tools is filled by the creation of an interpretable model for nonstationary neural networks based on linearized attention.

Quality: The robustness of the model is supported by numerous theoretical and experimental validations, including actual brain data.

Clarity: Although more illustrations would help the paper's complex theoretical content, the model's structure is explained in detail.

Significance: The work is very important since it offers a tool that may improve our comprehension of how neurons communicate, which could have ramifications for different applications in neuroscience.

**Weaknesses:**

Complexity: Some parts, especially the ones about the mathematical foundations, can be too difficult for a wider range of neuroscience readers.

Limitations of Interpretability: Although NetFormer can be interpreted in comparison to more conventional models, some results might be strengthened by further biological contextualization.

Evaluation Scope: The model performs well under partial observability, but it might require additional testing in a variety of scenarios found in real-world neuroscience datasets.

**Questions:**

Nonlinear Dynamics: Could the authors elaborate on whether using various non-linear activation functions in the simulation tests significantly alters the model's performance?

Scalability: How does the model adapt to bigger datasets, especially in real-world scenarios when partial observability is a problem?

Interpretability: Could the writers elaborate on how particular attention weights might be connected to recognized patterns of connection or brain phenomena?

---

> ### Author Response · Authors · 2024-11-22
> **Thanks for your constructive feedbacks**
>
> Thank you for your thoughtful review! We have revised our manuscript (with revisions marked in purple). We hope the revision and the following responses could address some of your concerns:
>
> - **Model performance across different nonlinear activations:** We performed a second set of connectivity-constrained simulation, where a sigmoid activation is used in place of the tanh activation. Then we used NetFormer, along with other baselines, to reconstruct connectivity at both individual-neuron level and cell-type level. The newly added results in Table 5 and Figure 11 in Appendix A7 show that NetFormer’s performance is consistent across both tanh and sigmoid activations. We would like to reiterate that activations with a sigmoidal shape, such as tanh and sigmoid, are of particular interest in the current neuroscience application. As we have also mentioned in Section 2, sigmoidal activations comply with the biological fact that the effect of one neuron on another can be either excitatory or inhibitory, but neither effect can be arbitrarily large.
>
> - **Adapting the model to bigger datasets where partial observability is a problem:** While we look forward to testing the scalability of our model on larger real-world datasets, the current dataset we used (Bugeon et al. 2022) is the largest public dataset that provides both activity and cell type information of simultaneously recorded neurons. Meanwhile, in the connectivity-constrained simulation (section 4, Fig 5c), we have assessed the robustness of NetFormer against partial observations at both individual-neuron level and cell-type level. We found that individual-neuron level connectivity recovery remains robust even with only half the neurons observed, and more robustness against partial observations could be achieved if we focus on cell-type level connectivity recovery.
>
> - **Connecting attention weights to recognized patterns of connection or brain phenomena:** We would like to first highlight that the attention weights that we recovered from the mouse cortex recording is in good agreement with the patch-clamp measured cell-type level connectivity strength (Campagnola et al., 2022). Moreover, we also observed changes in cell-type level connectivity estimated from attention weights that can depend on behavioral states, which is consistent with experimental reports in Fu et al. 2014. It is reported that VIP cells are more activated during locomotion, while SST cells are more inhibited, suggesting “a circuit in which VIP cells increase activity of neighboring excitatory cells by inhibiting their inhibitory input from SST cells” (Fu et al. 2014). In Fig 6d top row, NetFormer-inferred attention shows that the presynaptic VIP column (rightmost column) is darker (more inhibition) in the running state compared to the stationary states and the presynaptic SST column (second column from the right) is lighter (less inhibition). Moreover, we also evaluated the effectiveness of our approach on synthetic datasets which incorporate biological features, such as STDP (section 3.2), neuroscience task training (section 3.3), or cell-type level connectivity constraints (section 4).

---

### Official Review · Reviewer_9MUd · 2024-11-08

**Soundness:** 3
**Presentation:** 2
**Contribution:** 2
**Rating:** 8
**Confidence:** 4

**Summary:**

This work aims to recover the time-dependent connectivity structures in neural recordings using a transformer based model with a simple attention mechanism. The authors validate their approach on simulated RNNs and later apply to a published neural dataset.

**Strengths:**

- The simulations are mostly convincing (see below) of the approach's validity, providing compelling evidence for its use in biological experiments.

- To my knowledge, apart from [1], this is one of the first actionable use cases of transformer models in systems neuroscience.

**Weaknesses:**

I believe this work may benefit from at least one cycle of revisions, with new experiments added to strengthen the validity claims. Specifically, the weaknesses below are the main reason for my score, though a substantial revision can change my mind.

- Missing baseline: The authors claim that existing work mainly focus on stationary weight matrices and compare with respect to those. [2] is a relevant work that does not seem to be cited, which should also be one of the baselines.

- Missing validity experiments: I believe the validity of the approach should be further strengthened by considering task trained RNNs with short-term synaptic plasticity (claimed in Figs. 2 and 3).  See [3] for an example of how to validate with task-trained, not randomly connected, RNNs. [4] shows how to incorporate STSP into firing rate models and [5] does the same thing for the spiking network models.

- The experiment in Fig. 4 should use [7], which I believe is a more appropriate baseline.

**Questions:**

I believe I understood most of the work adequately, but I do have one question on Fig. 5: I am not exactly sure how fig. 5 is consistent with the findings from Fu et al. 2014. Could you explain a bit more, or maybe add a comparison panel to the figure highlighting this claim? Also, as a neuroscientist, to me, a large-scale dataset is one that has 10,000 to millions of neurons [6]. The dataset considered seems to have at most hundreds of neurons.

To address my concerns, you can perform the following revision:

1) Can you add [2] as a baseline to your experiments?

2) Can you please replace Fig. 2 (which can become supplementary) with a new figure that is based on task-trained RNNs? It is now well known that weights drawn randomly from a normal distribution may have some desirable properties, masking the true abilities of models like Netformer to capture the underlying connectivity [3].

3) Can you please replace Fig 3 with a task-trained RNN as well? If you wish, and this would be a spotlight level contribution, you can show the superiority of netformer in finding stsp patterns in both firing rate based [4] and spiking based [5] models.

4) For Fig. 4, I believe [7] would be a relevant baseline.

I believe, for a borderline score, the suggestions 1, 2 and 4 should be addressed.

Edit: The authors have performed substantial revisions to address my concerns 1; 2, and 4. Moreover, the addition of the mexican hat experiment makes this paper a solid contribution as a poster in my opinion.

[1] Kozachkov, Leo, Ksenia V. Kastanenka, and Dmitry Krotov. "Building transformers from neurons and astrocytes." Proceedings of the National Academy of Sciences 120.34 (2023): e2219150120.

[2] Pellegrino, Arthur, N. Alex Cayco Gajic, and Angus Chadwick. "Low tensor rank learning of neural dynamics." Advances in Neural Information Processing Systems 36 (2023): 11674-11702.

[3] Qian, William, et al. “Partial Observation Can Induce Mechanistic Mismatches in Data-Constrained Models of Neural Dynamics.” NeurIPS 2024, OpenReview, 25 Sept. 2024

[4] Masse, N. Y., Yang, G. R., Song, H. F., Wang, X. J., & Freedman, D. J. (2019). Circuit mechanisms for the maintenance and manipulation of information in working memory. Nature neuroscience, 22(7), 1159-1167.

[5] Mongillo, Gianluigi, Omri Barak, and Misha Tsodyks. "Synaptic theory of working memory." Science 319.5869 (2008): 1543-1546.

[6] Manley, Jason, et al. "Simultaneous, cortex-wide dynamics of up to 1 million neurons reveal unbounded scaling of dimensionality with neuron number." Neuron 112.10 (2024): 1694-1709

[7] Sani, Omid G., Bijan Pesaran, and Maryam M. Shanechi. "Dissociative and prioritized modeling of behaviorally relevant neural dynamics using recurrent neural networks." Nature Neuroscience (2024): 1-13.

---

> ### Author Response · Authors · 2024-11-22
> **Thanks for your constructive feedbacks**
>
> Thank you for your thoughtful review! We have revised our manuscript (with revisions marked in purple). We hope the revision and the following responses could address some of your concerns:
>
> - **Relating Fig 5 (now Fig 6) to Fu et al. 2014**: Thanks for the question. It is reported in Fu et al. 2014 that VIP cells are more activated during locomotion, while SST cells are more inhibited, suggesting “a circuit in which VIP cells increase activity of neighboring excitatory cells by inhibiting their inhibitory input from SST cells” (Fu et al. 2014). In Fig 6d top row, NetFormer-inferred attention shows that the presynaptic VIP column (rightmost column) is darker (more inhibition) in the running state compared to the stationary states, and the presynaptic SST column (second column from the right) is lighter (less inhibition).
>
> - **Dataset being limited compared to more recent ones (Manley et al. 2024)**: Thank you for introducing this valuable dataset. In this study, one of our main focuses is to investigate cell-type specific connectivity. As such, the current dataset (Bugeon et al. 2022) is central to our analysis, as it includes both simultaneously recorded neural activity and their cell type information. As pointed out in Bugeon et al. 2022, using calcium imaging in transgenic mice (as done in Manley et al. 2024) cannot identify neurons of different cell types simultaneously, which is crucial for validating our model. Given that fact, Bugeon et al. 2022 provides the largest public dataset that contains both activity and cell type information of simultaneously recorded neurons. As described in Appendix A.5.3, this dataset yields approximately 500 neurons per session, and for the subject we considered (SB025), a total of 2482 neurons were recorded across 6 sessions. These sessions have been employed to study NetFormer’s cross-session parameter sharing capacity in Fig 5d. We have also toned down the description of “large-scale” for the dataset size.
>
> - **Adding LtrRNN (Pellegrino et al., 2023) as a baseline**: Thank you for the suggestion. We have added LtrRNN (section 4 and Fig 6). We found that the inferred connectivity from LtrRNN is not in good agreement with the cell-type level patch-clamp measured ground truth (Fig 6 and Appendix A8). We further observed that compared to weights inferred by LtrRNN, PCA on NetFormer-inferred weights yields a better separation between running and two stationary states. We also want to comment that NetFormer is more efficient than LtrRNN: for a given neuronal population, NetFormer uses shared parameters across time/trials, while LtrRNN trains trial-specific parameters, leading to an increase in parameters as the number of trials grows.
>
> - **New figure on task-trained RNNs:** Thanks for the inspiring question. We added new experiments on applying NetFormer to infer the connectivity between hidden units of task-trained RNNs (see section 3.3 and Fig 4). We considered three representative tasks from the NeuroGym toolkit [1], which involve integration of evidence over time, and/or maintaining working memory to make decisions after a delay period. Compared to a linear model, we observed that NetFormer achieves higher accuracy in both dynamics prediction and connectivity recovery (see Fig 4 and Table 4 in Appendix 4.2). While we agree that extending our experiments to task-trained RNNs with STSP is a valuable future direction, we believe that the two core concepts there have been verified: The new task-trained RNN results show that NetFormer can capture task-driven connectivity patterns, and our previous result on STDP shows that NetFormer can capture synaptic modification.

---

> ### Author Response · Authors · 2024-11-22
>
> - **Adding Sani et al. 2024 as a baseline:** After a careful reading of Sani et al. 2024, we found their focus to be very different from ours, and thus might not be a suitable baseline for comparison. While their work emphasizes capturing behaviorally relevant neural activity patterns, our work focuses on modeling nonstationary connectivity between neurons. We realized that the confusion may arise from our use of behavioral features in Fig 6. To clarify, during training, our model works solely with neural activity, without access to any behavioral information. Specifically, our model is trained to make next-step neural activity predictions as accurately as possible. After training, we compared the time-varying connectivity inferred by our model to the behavioral information, and found that the *unsupervised* patterns emerging from the time-varying connectivity are actually behaviorally meaningful. In contrast, Sani et al. 2024 uses both neural and behavioral activity as model inputs and prediction targets for training. Their focus is not on recovering connectivity between neurons, but on modeling the neural dynamics that give rise to the observed behavior. As a result, they discovered relationships between neural activity and behavior in a *supervised* way, and their model is less interpretable in terms of neuron-to-neuron connectivity.
>
> **Reference**
>
> [1] Molano-Mazon, M., Barbosa, J., Pastor-Ciurana, J., Fradera, M., Zhang, R. Y., Forest, J., ... & Yang, G. R. (2022). NeuroGym: An open resource for developing and sharing neuroscience tasks.

---

> > ### Comment · Reviewer_9MUd · 2024-11-22
> >
> > Dear Authors,
> >
> > I think your responses constitute a master class in how to respond to critical reviews and improve your own work on top of them. I think my concerns 1, 2 and 4 are addressed (specifically, the addition of rSLDS results are just as desirable as comparing to [7]). As promised, I will raise my score to a borderline accept.
> >
> > Now, since your added results are quite interesting and must have taken you a good amount of time and energy, I am willing to support a stronger acceptance IF you commit to performing the Mexican hat experiment from Das and Fiete 2020 (also requested by another reviewer). The results do not matter, even if NetFormer ends up inferring spurious connectivity, as long as you are able to quantify it the way it was done in Das and Fiete, that would be a clear acceptance in my opinion. I am not asking you to do these experiments in 5 days (if you can I would appreciate it) I only ask you to commit in order to increase my score further. I believe this would be a crucial addition that should be in this paper and not in some future work.

---

> > > ### Author Response · Authors · 2024-11-22
> > >
> > > Dear Reviewer,
> > >
> > > Thank you for your constructive feedback, which helps us further improve our work. We appreciate your recognition of our efforts and the adjustment to the score. We plan to work on the Mexican Hat experiment from Das and Fiete (2020) and will provide updates as time permits.

---

> > > > ### Comment · Reviewer_9MUd · 2024-11-23
> > > >
> > > > Dear Authors,
> > > >
> > > > Please do update me if you can, I am certainly very excited. I updated my score to support an acceptance. In my opinion, this work, after promised revisions, meets the bar for publication at ICLR

---

> > > > > ### Author Response · Authors · 2024-11-27
> > > > >
> > > > > Dear Reviewer,
> > > > >
> > > > > Thank you for the suggestion. In Appendix 9, we conducted two experiments on simulated neuron populations with strong recurrent connections: one generating continuous activity and the other generating spiking activity, which follows Das & Fiete (2020) more closely but requires modifying NetFormer’s objective function. We experimented with different values of recurrent strength in the both simulated networks. Adjusting recurrent strength alters the network’s regime, leading to stronger or weaker levels of spurious correlations. In both simulations, NetFormer shows some resilience to spurious correlations, while can also struggle when the recurrent weights are too strong or too weak, similar to other methods studied in Das & Fiete (2020).

---

### Author Response · Authors · 2024-11-22
**Thanks for your constructive feedbacks**

We thank all reviewers for their thoughtful suggestions. We are much encouraged to see that our work has been deemed “well-conceived, bridging gaps left by traditional static models” (ipsn) and our novel model “is one of the first actionable use cases of transformer models in systems neuroscience”(9MUd), “does indeed address an important question” (ovPw), and “a genuine gap in the present neuroscience tools is filled by the creation of an interpretable model for nonstationary neural networks based on linearized attention” (HFca).

Based on all reviewers’ comments, we have carefully revised our manuscript (with revisions marked in purple). We have uploaded our code in Supplementary Material, and we will make it publicly available. We kindly invite all reviewers to review our revised manuscript, and we also provide an overview of the major updates below:

* **Task-driven population activity simulation (section 3.3, Fig 4).** As suggested by reviewer 9MUd, we further tested whether NetFormer can identify task-driven connectivity patterns in neural populations. Specifically, on three representative neuroscience tasks from the NeuroGym toolkit [1], we trained recurrent neural networks (RNNs) to perform the tasks, and applied NetFormer to learn the connectivity of trained RNNs from the activity of their recurrent units. NetFormer achieves higher accuracy in both dynamics prediction and connectivity recovery compared to linear models (Table 4 in Appendix A.4.2, also attached below).

     |   | NetFormer (MSE)  | NetFormer ($R^2$) | Linear (MSE)   | Linear ($R^2$) | NetFormer (Spearman) | NetFormer (Pearson) | Linear (Spearman) | Linear (Pearson) |
     |-------------|------------------|-------------------|----------------|----------------|-----------------------|---------------------|-------------------|------------------|
     | **a**       | **0.000** ± 0.000 | **0.998** ± 0.001 | 0.014          | 0.920          | **0.694** ± 0.163     | **0.760** ± 0.120   | 0.566            | 0.548
     |
     | **b**       | **0.010** ± 0.000 | **0.972** ± 0.001 | 0.013          | 0.960          | **0.622** ± 0.006     | **0.722** ± 0.004   | 0.518            | 0.553
     |
     | **c**       | **0.001** ± 0.000 | **0.997** ± 0.001 | 0.034          | 0.897          | **0.811** ± 0.011     | 0.633 ± 0.105       | 0.626            | **0.650**
     |

* **More advanced and well-recognized baselines designed for modeling nonstationary connectivity (section 4, Fig 6).** As suggested by reviewer 9MUd, ipsn, and ovPw, we further introduced two baselines that can account for trial- and (state-) varying connectivity: low-tensor-rank RNN (LtrRNN, [2]) and autoregressive Hidden Markov Model (AR-HMM, [3, 4]). LtrRNN models trial-varying connectivity using a low rank tensor $\mathbf{W} \in \mathbb{R}^{N \times N \times K}$, where $K$ denotes the number of trials. AR-HMM assumes that there are discrete latent states switching underlying the observed activity, and each state admits unique dynamics through a different connectivity matrix. We found that NetFormer is able to capture connectivity changes between behavioral states more clearly. Moreover, compared to LtrRNN and AR-HMM, connectivity inferred from NetFormer's attention is in better agreement with the cell-type level patch-clamp measurements. (Table 6 in Appendix A.8, also attached below).

    | Context               | Condition                  | Metric     | NetFormer                | LtrRNN                   | AR-HMM                   |
    |-----------------------|----------------------------|------------|--------------------------|--------------------------|--------------------------|
    | **Running**           | K x K                     | Pearson    | **0.591** ± 0.204        | 0.007 ± 0.182            | 0.199 ± 0.027            |
    |                       |                            | Spearman   | **0.601** ± 0.209        | -0.036 ± 0.189           | 0.274 ± 0.048            |
    | **Stationary Desync** | K x K                     | Pearson    | **0.662** ± 0.176        | -0.003 ± 0.181           | -0.320 ± 0.012           |
    |                       |                            | Spearman   | **0.723** ± 0.173        | -0.057 ± 0.174           | -0.206 ± 0.017           |
    | **Stationary Sync**   | K x K                     | Pearson    | **0.713** ± 0.148        | 0.000 ± 0.186            | 0.145 ± 0.041            |
    |                       |                            | Spearman   | **0.767** ± 0.151        | 0.069 ± 0.175            | 0.151 ± 0.009            |

* **Visualization of NetFormer’s dynamics prediction on neural data (Fig 13, Appendix A.8).** As suggested by reviewer ipsn and ovPw, we have added visualizations of true and predicted neural activity as well as connectivity in different behavioral states in Fig 13, Appendix A.8.

---

### Author Response · Authors · 2024-11-22

**References**

[1] Molano-Mazon, M., Barbosa, J., Pastor-Ciurana, J., Fradera, M., Zhang, R. Y., Forest, J., ... & Yang, G. R. (2022). NeuroGym: An open resource for developing and sharing neuroscience tasks.

[2] Pellegrino, A., Cayco Gajic, N. A., & Chadwick, A. (2023). Low tensor rank learning of neural dynamics. Advances in Neural Information Processing Systems, 36, 11674-11702.

[3] Fox, E., Sudderth, E., Jordan, M., & Willsky, A. (2008). Nonparametric Bayesian learning of switching linear dynamical systems. Advances in neural information processing systems, 21.

[4] Linderman, S., Johnson, M., Miller, A., Adams, R., Blei, D., & Paninski, L. (2017, April). Bayesian learning and inference in recurrent switching linear dynamical systems. In Artificial intelligence and statistics (pp. 914-922). PMLR.

---

### Author Response · Authors · 2024-11-27
**Thank you for your constructive feedback**

We appreciate all reviewers’ feedback and comments on our work; our manuscript has benefited substantially from the many valuable suggestions we have received during this rebuttal period. In the latest revision, we have included new experiments in Appendix 9 on applying NetFormer to networks showing spurious correlations. In summary, we found that NetFormer shows some resilience to spurious correlations, while can also struggle when the recurrent connections are too strong or too weak, similar to other methods studied in Das & Fiete (2020). Nevertheless, since this is a well-recognized challenge for models aiming to infer connectivity from fitting activity, we see it as an opportunity for future improvement. Our latest revision is marked in red, and our code for these experiments are also attached in supplementary materials. We kindly invite all reviewers to review our revised manuscript.

---

### Meta-Review · Area_Chair_vB4D · 2024-12-19

**Metareview:**

The authors present a novel approach for reconstructing time-varying neural connectivity from data based on a transformer-like self-attention model, and show good results on model and real data. The authors were unanimous that it is a good paper worth publishing, and I agree. As this is a topic which has been widely studied by the ML community for years, it might be a good idea to include some more baselines, such as classic work on state space models (in the sense of HMMs and Kalman Filters with GLM observation models, not the more modern sense of Mamba, S5 etc) e.g. Pfau, Pnevmatikakis and Paninski (NeurIPS 2013), but it is not necessary for the paper to be accepted.

**Additional Comments On Reviewer Discussion:**

The reviewers all engaged well with the authors and were convinced that their major objections were addressed.

---

### Decision · Program_Chairs · 2025-01-22

Accept (Spotlight)